

# Objective evaluation of surface- and satellite-driven CO₂ atmospheric inversions

Frédéric Chevallier[1], Marine Remaud[1], Christopher W. O'Dell[2], David Baker[2], Philippe Peylin[1], Anne Cozic[1]

5    [1]Laboratoire des Sciences du Climat et de l'Environnement, LSCE/IPSL, CEA-CNRS-UVSQ, Université Paris-Saclay, F-91198 Gif-sur-Yvette, France
[2]Cooperative Institute for Research in the Atmosphere, Colorado State University, Fort Collins, CO, USA

*Correspondence to*: Frédéric Chevallier (frederic.chevallier@lsce.ipsl.fr)





**Abstract.** We study an ensemble of six multi-year global Bayesian $CO_2$ atmospheric inversions that vary in terms of assimilated observations (either column retrievals from one of two satellites or surface air sample measurements) and transport model. The time series of inferred annual fluxes are first compared with each other at various spatial scales. We then objectively evaluate the small inversion ensemble based on a large dataset of accurate aircraft measurements in the free troposphere over the globe, that are independent from all assimilated data. The measured variables are connected with the inferred fluxes through mass-conserving transport in the global atmosphere and are part of the inversion results. Large-scale annual fluxes estimated from the bias-corrected land retrievals of the second Orbiting Carbon Observatory (OCO-2) differ from the prior fluxes much, but are similar to the fluxes estimated from the surface network within the uncertainty of these surface-based estimates. The OCO-2- and surface-based inversions have similar performance when projected in the space of the aircraft data, but relative strengths and weaknesses of the two flux estimates vary within the Northern and Tropical parts of the continents. The verification data also suggests that the more complex and more recent transport model does not improve the inversion skill. In contrast, the inversion using bias-corrected retrievals from the Greenhouse Gases Observing Satellite (GOSAT) or, to a larger extent, a non-Bayesian inversion that simply adjusts a recent bottom-up flux estimate with the annual growth rate diagnosed from marine surface measurements, estimate much different fluxes and fit the aircraft data less. Our study highlights a way to rate global atmospheric inversions. It suggests that some satellite retrievals can now provide inversion results that are, despite their uncertainty, comparable in credibility to traditional inversions using the accurate but sparse surface network and that are therefore complementary for studies of the global carbon budget.

## 1. Introduction

Carbon dioxide ($CO_2$) is increasingly monitored in the global atmosphere due to its important role in climate change. For example, NOAA's GlobalView Plus Observation Package (ObsPack, Cooperative Global Atmospheric Data Integration Project, 2018) archives high-quality measurements made at the surface or from aircraft by various institutes. Despite occasional budget difficulties (Houweling et al., 2012), the number of collected data points has exponentially increased over the years, with, in reference to 1980, six times more measurements in 2000 and 100 times more measurements in 2017. In addition, the ground-based Total Carbon Column Observation Network of column retrievals (TCCON, Wunch et al., 2011) is less than 15 years old but already operates about 30 sites over the globe. Other measurements, like the recent AirCore technique that samples air in freefall tubes (Karion et al., 2010) or the COllaborative Carbon Column Observing Network (COCCON, Frey et al. 2018), have also emerged in the past decade. Most remarkably, the number of spectrometers designed to monitor the $CO_2$ column from space has grown from one in 2002 to six at the end of 2018 (Crisp et al., 2018). The primary motivation for this increase of $CO_2$ observations has been to further our understanding of the global surface fluxes of carbon, with the additional help of meteorological data (e.g., Bolin and Keeling, 1963; WMO, 2018). This is done in practice by inversion of atmospheric transport models within a Bayesian framework (e.g., Peylin et al., 2013). Scientists have urged caution when interpreting this growing amount of data because the uncertainty of the available meteorological information was identified early as a critical limitation on the exploitable measurement information. This limitation motivated the creation of the international Atmospheric Tracer Transport Model Intercomparison project 25 years ago (TransCom, Law et al., 1996) and is still relevant today (Schuh et al., 2018). Adequate representation of the various error statistics involved in the Bayesian estimation remains a challenge (e.g., Bocquet et al., 2011). In addition, column retrievals, made from measured radiances from space or on the ground after complex processing, cannot fundamentally be calibrated relative to WMO-traceable standards, in contrast to surface measurements like those in ObsPack GlobalView Plus. Indeed, systematic errors in the retrievals at the sub-µmol/mol level ($10^{-6}$ mol/mol, abbreviated as part per million, ppm) are enough to affect the flux estimation (Chevallier et al. 2007), but the current TCCON retrievals that serve as the best reference for column retrievals with global coverage, have commensurate offset uncertainties (Wunch et al., 2015).

A given inversion configuration is made of one or several observation types, a transport model and a few statistical models. Many of them seem reasonable. Though model disagreement has been reduced over the last couple of decades, current inversion results show an unacceptably large spread, even for zonal averages (e.g., Le Quéré et al., 2018). This study aims at evaluating whether simple measures of quality based on airborne measurements in the free troposphere can distinguish between six inversion configurations. These inversion configurations differ in the assimilated data and in the transport model. The assimilated data are either surface measurements in ObsPack and related databases, retrievals from the Greenhouse Gases Observing Satellite (GOSAT) or the second Orbiting Carbon Observatory (OCO-2). The transport models are two versions of the atmospheric general circulation model of the Laboratoire de Météorologie Dynamique (LMDz, Hourdin et al., 2013) nudged towards analysed meteorological variables. The Bayesian inversion system from the Copernicus Atmosphere Monitoring service (CAMS, https://atmosphere.copernicus.eu/, Chevallier et al., 2005) is used in all six inversions. We use a "poor man's inversion" (Chevallier et al., 2009) based on recent bottom-up fluxes and on the global annual atmospheric growth





rate estimated from the average of marine surface measurements (Conway et al., 2014) to define a baseline for the skill of each Bayesian inversion result.

Our use of airborne measurements in the free troposphere as verification data is motivated by their frequent, WMO-traceable calibration, their independence from all data assimilated here (including the measurements in the boundary layer) and their
spatial distribution that samples all oceans and continents. Arguably they are the only $CO_2$ dataset that possesses all of three qualities.

In the following, data and models are described in Section 2, while Section 3 presents the various results. They are discussed in Section 4. Section 5 concludes the study.

## 2.   Model, system and data
### 2.1.  Transport models

LMDz is the atmospheric component of the Earth system model of Institut Pierre-Simon-Laplace (Dufresne et al., 2013) which has been contributing to the recent versions of the Climate Model Intercomparison Project (CMIP) established by the World Climate Research Programme (https://cmip.llnl.gov/). Here, we use its off-line version (Hourdin et al. 2006) to simulate the transport of $CO_2$. The off-line LMDz model reads a frozen archive of 3-hourly-mean meteorological data pre-computed by the
full LMDz so that it only needs to simulate large-scale advection and subgrid transport processes (i.e. deep convection and boundary-layer turbulence). LMDz is nudged towards 6-hourly analysed meteorological variables, here either ERA-Interim (Dee et al., 2011) or ERA-5 (https://www.ecmwf.int/en/forecasts/datasets/archive-datasets/reanalysis-datasets/era5, access 31 January 2019) with a relaxation time of 3 hours. On-line and Off-line models are consistently run at the same spatial resolution in order to avoid any challenging interpolation of the air mass fluxes for the subgrid processes: here 39 eta-pressure layers
between the surface and around 80 km above sea level, and 96×96 grid-points, i.e. a horizontal resolution of 1.89°×3.75° in longitude. This configuration discretizes the 2-7 km above sea level region of the atmosphere, that will be a major focus in the following, into 6 to 10 layers, depending on local orography.

We use two physical formulations of LMDz, called 5A (in code identification number 1649) and 6A (in code identification number 3353), as described by Remaud et al. (2018, and references therein). The gap between the two versions represents
about six years of development from the LMDz team and includes, e.g., a complete revision of radiation, the introduction of the thermodynamical effect of ice and changes in the subgrid-scale parameterizations (convection, boundary layer dynamics) and in the land surface processes. For version 5A, horizontal winds are nudged towards ERA-Interim, but we use the new ERA-5 for LMDz6A. Therefore, the differences between the two versions cannot be exclusively attributed to subgrid-scale processes, since boundary variables (nudging files and land processes) differ as well.

### 2.2.  Inversion system

LMDz is embedded within the CAMS $CO_2$ inversion system. This system minimizes a Bayesian cost function to optimize the grid-cell eight-day surface fluxes (with a distinction between local night-time fluxes and daytime fluxes, but without fossil fuel emissions, that are prescribed) and the initial state of $CO_2$. To do so, it assimilates a series of $CO_2$ observations over a given time window within the LMDz model. The minimization approach is called 'variational' because it explicitly computes
the gradient of the cost function using the adjoint code of LMDz. Prior information about the surface fluxes is provided to the Bayesian system by a combination of climatologies and other types of measurement-driven flux estimates (e.g., Emission Database for Global Atmospheric Research version 4.3.2, Crippa et al, 2016, scaled globally and annually from Le Quéré et al., 2018, for the fossil fuel emissions or Landschützer et al., 2017, for the ocean fluxes). Details can be found in Chevallier (2018). Of special interest here is the fact that, when integrated over a calendar year, prior natural fluxes are zero over all land
grid points: this implies that the interannual variability of the inferred annual-mean of terrestrial vegetation fluxes is generated by the assimilated observations only. Over a full year, the total 1-sigma uncertainty (resulting from assigned error variances that vary in space and time, and from assigned temporal and spatial error correlations) for these prior land fluxes amounts to about 3.0 GtC·a$^{-1}$. The error statistics for the open ocean correspond to a global air-sea flux uncertainty about 0.5 GtC·a$^{-1}$.

The assimilation window is either 19 years for the surface measurements (from January 2000 until October 2018), eight years
for the GOSAT retrievals (from January 2009 until December 2016) or four years for the OCO-2 retrievals (from September 2014 until July 2018).

### 2.3.  Assimilated observations





All assimilated observations are dry air mole fraction of $CO_2$.

Assimilated surface air sample measurements have been selected from four large ongoing databases of atmospheric $CO_2$ measurements: (i) NOAA's ObsPack (Cooperative Global Atmospheric Data Integration Project, 2018, and CarbonTracker Team, 2018), (ii) the World Data Centre for Greenhouse Gases archive (WDCGG, https://gaw.kishou.go.jp/), (iii) the Réseau
Atmosphérique de Mesure des Composés à Effet de Serre database (RAMCES, http://www.lsce.ipsl.fr/), and (iv) the Integrated Carbon Observation System- Atmospheric Thematic Center (ICOS-ATC, https://icos-atc.lsce.ipsl.fr/). The list of selected sites is given by Chevallier (2018). Each dataset provides at least five years of measurements. The error variances assigned to these measurements in the inversion system correspond to transport modelling uncertainty (analytical measurement uncertainty of in situ $CO_2$ data is a negligible component) and are computed as the variance of the high frequency variability of the de-
seasonalized and de-trended $CO_2$ time series of the daily-mean measurements at each site. These variances are then inflated in order to give the same weight to each measurement day at a given location.

GOSAT was launched in January 2009, as a joint project of Japan Aerospace Exploration Agency (JAXA), NIES (National Institute of Environmental Studies) and Japan's Ministry of the Environment (MOE) (Kuze et al., 2009). OCO-2 is a NASA satellite that was launched in July 2014 (Eldering et al. 2017). Both satellites still collect scientific data today. They orbit
around the Earth from pole to pole with a local crossing time at the Equator in the early local afternoon. Each carry a spectrometer that measures the sunlight reflected by the Earth and its atmosphere in the near-infrared/ shortwave infrared spectral regions, with high spectral resolution ($>\sim 20,000$) such that individual gas absorption lines are resolved. OCO-2 provides spatially dense data with a narrow swath and with footprints of a few $km^2$, while GOSAT provides coarser-resolution data (100 $km^2$ at nadir) with low spatial density. Various algorithms have been developed to retrieve the column-average dry
air-mole fraction of $CO_2$ in the atmosphere ($XCO_2$) from the measured radiance spectrums. For GOSAT, we use bias-corrected $XCO_2$ retrievals from product OCO Full Physics (OCFP) v7.1 made by the University of Leicester and available from the Copernicus Climate Change Service for the period April 2009 – December 2016 (https://climate.copernicus.eu/). For OCO-2, we use NASA's Atmospheric $CO_2$ Observations from Space (ACOS) bias-corrected retrievals, version 9 (Kiel et al., 2018; O'Dell et al., 2018) from September 2014 until July 2018. In both cases, a previous release of the CAMS surface-based
inversion contributed to the retrieval official bias-correction to some extent. We neglect this dependency in the following because other reference data are used that reduce the weight of the CAMS inversion (e.g., TCCON), and because the bias-correction schemes rely on 2 to 5 time- and space- invariant parameters only, with internal retrieval variables (e.g., the retrieved vertical $CO_2$ gradient between the surface and the free troposphere) as predictors. We do not tune the official retrieval bias-corrections. To reduce data volume without loss of information at the scale of a global model, OCO-2 retrievals have been
averaged in 10-s bins for the Model Intercomparison Project (MIP) of OCO-2, as described in Crowell et al. (2019), and we use them in this form. The retrieval averaging kernels, prior profiles and Bayesian uncertainty are accounted for in the assimilation of both types of satellite retrievals. For OCO-2 retrievals, we also use the transport uncertainty term that is provided by the OCO-2 MIP (Crowell et al., 2019).

We only consider "good" retrievals as identified by variable *xco2_quality_flag* of each product. Both land and ocean data are
used for GOSAT. GOSAT data over ocean have matured in the ~10 years since they were first produced, and have reached a point where they appear to have smaller biases than over land (Zhou et al., 2016). Their direct inclusion in inversions also appears to be beneficial (Deng et al., 2016). However, though the ocean biases in OCO-2 have been substantially reduced since the initial version 7 (O'Dell et al. 2018), initial inversion tests using OCO-2 ocean observations still produced highly unrealistic results and are hence left out of this work. As for GOSAT, this situation may change in time and OCO-2 ocean
data could be beneficial in future inversion set-ups. Despite the exclusion of ocean retrievals and the 10 s averaging, there are still 65% more OCO-2 retrievals than GOSAT retrievals assimilated on average per month.

### 2.4. Verification observations

We use some specific measurements of the dry air mole fraction of $CO_2$ as verification data. They are aircraft measurements in the free troposphere made between July 2009 and December 2017 and archived in different ObsPacks (Cooperative Global
Atmospheric Data Integration Project, 2018, and NOAA Carbon Cycle Group ObsPack Team, 2018). Table 1 lists the various aircraft measurement sites, campaigns or programs. For simplicity, all sites, campaigns or programs will be referred to as "programs" in the following. All measurements have been calibrated to the WMO $CO_2$ X2007 scale or to the NIES 09 $CO_2$ scale to better than 0.1 ppm (e.g., Machida et al., 2008; Sweeney et al., 2015). We note that no aircraft data is assimilated here (Section 2.3).

We define the free troposphere as the altitudes comprised between 2 and 7 km above sea level. We avoid data below 2 km because (i) local anthropogenic emissions affect many aircraft measurements there, and (ii) some of the aircraft flew in the





vicinity of measurement sites that have been used in the surface-based inversions. We avoid data above 7 km because the measurement variations (and the flux regional signal) are much reduced there. A few outliers for which the difference between model and observation is larger than 40 ppm are rejected.

We define two periods for the following statistical computations. They are based on the availability of the satellite retrievals and of the aircraft data in the databases used here: a "GOSAT period" from July 2009 until September 2016 and an "OCO-2 period" from December 2014 until December 2017. Note that they overlap and that there is a minimum of three months between the temporal bounds of the verification data and the temporal bounds of the assimilated data in order to account for inversion spin-up and spin-down. Figure 1 shows the geographical location of the verification data for the two periods.

### 2.5. Poor man's inversion

In order to put the differences between inversion simulations and aircraft measurements in perspective, we compare them to an inversion that only assimilated the annual global growth rate of $CO_2$. This baseline, called "the poor man's inversion" by Chevallier et al. (2009), adjusts prior natural fluxes over land in order to fit the annual trend of globally-averaged marine measurements (http://www.esrl.noaa.gov/gmd/ccgg/trends/, access 10 January 2019) multiplied by a conversion factor (2.086 GtC·ppm$^{-1}$, from Prather, 2012) when combined with prior ocean and fossil fuel fluxes. The correction to the natural land

fluxes is made proportional to the prior error standard deviations assigned within a given inversion system. In the case of the CAMS system here, the prior error standard deviations are themselves proportional to a climatology of heterotrophic respiration fluxes simulated by a vegetation model, with a ceiling of 4 gC·m$^{-2}$·day$^{-1}$. This simple approach is not Bayesian because prior error correlations are ignored, but it still allows transport models to fit atmospheric data with less bias than its prior fluxes because it closes the carbon budget in a plausible way.

Over the ocean and for the fossil fuel emissions, we choose the same prior fluxes as for the six Bayesian inversions (Landschützer et al., 2017, Crippa et al, 2016, Le Quéré et al., 2018, see Section 2.2). However, we choose more informed natural fluxes over land than for the Bayesian inversions: rather than letting the inversion fully free to locate the annual land sinks (see Section 2.2), we take a simulation of a dynamic global vegetation model that accounts for land-use, climate and $CO_2$ history (simulation ORCHIDEE-Trunk in Le Quéré et al., 2018). When multiplied by 2.086 GtC·ppm$^{-1}$, this combination of

prior fluxes already fits the annual trend of globally-averaged marine measurements with a root-mean-square difference of 0.3 ppm·a$^{-1}$, By construction, the poor man's adjustment brings these annual global differences to zero.

For the comparison of the poor man's inversion with aircraft measurements, we use LMDz5A. We start the poor man's simulation on 1 January 2000 from a 3D prior initial state of $CO_2$. We then add an offset to the simulation so that its mean bias with respect to NOAA's surface measurements at South Pole Observatory (Cooperative Global Atmospheric Data Integration

Project, 2018) over the 2010-2017 period is zero. This offset addresses the uncertainty of the initial state and the uncertainty of the 2.086 GtC·ppm$^{-1}$ conversion factor.

## 3.   Results
### 3.1. Principle

We build an ensemble of six Bayesian inversions using the inversion system of Section 2.2, the two transport model versions

of Section 2.1, and the three observation datasets of Section 2.3. The assimilation periods differ (Section 2.2), but the prior fluxes and the prior error model are the same. For each inversion, the posterior model simulation statistically fits its own assimilated data well within their 1-sigma uncertainty. Note that the surface-based inversion with LMDz5A is exactly the $CO_2$ inversion product 18r1 of CAMS that was released in November 2018 (http://atmosphere.copernicus.eu/). In the figures, we will refer to the surface-based inversions by the generic name "SURF" for simplicity.

We first present the carbon budget estimates. We choose to look at fluxes at the annual scale only, knowing that over land, the inferred interannual variability is completely driven by the assimilated observations (because prior natural fluxes over land are zero on annual average for the Bayesian inversions, see Section 2.2). As we will see, it is relatively large. Except at the global scale, capturing the interannual variability well is particularly challenging because its estimation accumulates all errors made throughout the seasonal cycle.

Then we compare the inversion performance vis-à-vis the aircraft measurements of Section 2.4, to the performance of the poor man's inversion of Section 2.5. This comparison is made for two periods (Section 2.4). For each of them, we will only consider the inversions that cover the window completely, which means that the GOSAT-based (or OCO-2-based) inversions will not be used in the results for the "OCO-2 period" (or "GOSAT period"). The projection of the inversion fluxes onto the space of



the aircraft-measured variables (mole fractions) is made by the same LMDz model version that was used in the inversion. Doing this, we are consistent with the way the inversion system distributes the well-constrained total mass of carbon in the atmosphere and we avoid error compensations between the version used in the assimilation and the one used in the evaluation. The model is directly sampled at measurement time and space, without any interpolation.

## 3.2. Annual budgets

The time series of the annual natural carbon budgets at several very broad scales are displayed in Figure 2 for the period between 2004 and 2017: the globe, the northern or southern extra-Tropics, and the Tropics with lands and oceans either separated or combined. At this scale, the influence of the transport model version is hardly distinguishable (coloured solid lines vs. coloured dashed lines). The poor man's inversion (black dashed lines) locates the land sink mostly in the northern
extra-Tropics but also in the Tropics (consistent with its prior information shown in Fig. 8 of Le Quéré et al., 2018), while the six Bayesian inversions put it more in the northern extra-Tropics (starting from a null prior on annual average). All approaches converge towards near-neutral southern extra-Tropical lands (that represent a relatively small surface area). Over the oceans, the surface-based inversions vary little from the prior (which is equal to the poor man's estimate there), but the GOSAT-based inversions reduce the ocean sink by about 0.5 GtC·a$^{-1}$ in 2015; the OCO-2-based inversions increase it by up to 1 GtC·a$^{-1}$. We
recall that years 2015 and 2016 correspond to a strong El Niño event associated with a large $CO_2$ growth rate (e.g., Mahli et al., 2018 and references therein). The GOSAT inversions seem to underestimate the beginning of this anomaly (see the top row of Figure 2), and to attribute it to the southern extra-Tropical oceans rather than to the Tropical lands like the other inversions. OCO-2-based fluxes are close to the surface-based fluxes, except for the increased ocean sink (which appears to be regularly spread between the three bands). The OCO-2-based and surface-based growth rates are very similar, but do not
fully overlap with the poor-man fluxes because they do not fully agree with NOAA's estimates, in particular in 2016 when they diagnose a smaller rate (by 0.25 ppm·a$^{-1}$ if we use the 2.086 GtC·ppm$^{-1}$ conversion factor).

Figure 3 and Figure 4 focus on the Bayesian inversion results at the scale of the 22 regions of the TransCom 3 experiment (Gurney et al., 2002): 11 regions over land and 11 regions over the oceans that together tile the whole globe. At this scale, the impact of the choice of the LMDz version appears: LMDz6A induces slightly less year-to-year variability for the surface-
based inversion for some years (see the 2010s for region Europe, the last couple of years for region Eurasian Temperate, or the full time series for region North Atlantic Temperate), and the two model versions can yield different baselines (see regions North and South American Temperate, or the three Atlantic regions). The two GOSAT-based inversions show larger year-to-year variability than the other ones. The OCO-2-based inversions broadly agree with the surface-based inversions for the temporal variability of the fluxes in most regions (North American Boreal, Southern Africa, Eurasian Boreal, Tropical Asia,
Europe) but there are noticeable differences in the North American Temperate, South American Tropical and Temperate, and Australia regions. While being clearly distinct from the inversion prior fluxes (that are zero on annual average over land), and from the GOSAT-based fluxes, we note the agreement of the two OCO-2-based inversions with the 6A SURF inversion and the poor man's fluxes (that are informed by an up-to-date bottom-up simulation) in the two boreal regions, despite the lack of OCO-2 data there during half of the year as a consequence of insufficient insolation (see, e.g., Deng et al., 2014). The main
differences between inversions OCO-2 and SURF over the ocean are regions North Pacific Temperate and Southern Ocean.

Figure 5 compares the difference between fluxes estimated by assimilating either OCO-2-or the surface data within LMDz6A, to the posterior uncertainty diagnosed from the Bayesian system (Chevallier et al. 2007) for the surface-based inversion. For all regions discussed so far, this difference is usually within the Bayesian uncertainty standard deviation (but reaches up to 2.6 times this quantity in Northern Africa for 2015), which means that the difference between the two flux estimates at this scale
is mostly not statistically significant.

Figure 6 further zooms in to the pixel-scale for year 2015, a year that is common between all inversions. Only the LMDz5A results for the satellite-based inversions are shown. For the two surface-based inversions, the change of transport model leaves the flux patterns generally unchanged but slightly modulates their amplitude. In contrast, the two satellite-based inversions show more differences in the flux pattern. They suggest large flux gradients in southern Africa and South America: similar
together in Africa, with a large sink in the tropical evergreen forests and large sources around; different in America with a source over the tropical evergreen forests for GOSAT and over a northeast corner for OCO-2. The broad flux patterns in the lands of the Northern Hemisphere are similar between the four maps, but OCO-2 has flux gradients closer to SURF than to GOSAT in America while the opposite is seen in South-east Asia. The Tropical ocean outgassing region reduces with OCO-2 and expands to the south with GOSAT.

## 3.3. Differences with aircraft data





Figure 7 presents the statistics of model-minus-measurement differences per measurement program for the GOSAT period. Note that the data number varies by several orders of magnitude among the programs: there are a few hundreds of samples for most of the 37 programs, but a few thousands for CALNEX2010, KORUS-AQ, ORCAS, SGP and ATom, a few 10,000s for ACT, DC3, DISCOVER-AQ, GSFC, HIPPO, SEAC4RS, and SONGNEX2015, and 900,000 for CONTRAIL. Obviously,

many measurements may fit into a single time-space block of the global transport model. We will only discuss bias differences larger than 0.15 ppm (i.e. above the calibration uncertainty of the aircraft data, see Section 2.4) and that are statistically significant at the 0.05 level, as reported on the figure. The computation of the significance level is made with an unpaired $t$-test when comparing inversion results that assimilated different data (we assume that changing the assimilated data makes the inversion results independent), and with a paired $t$-test when comparing inversion results that assimilated the same data (we

assume that inversion results in which only the transport model varies are dependent). In practice, changing the independency assumption only affects the detail of the significance-level results, but not the overall picture.

Comparing solid and dotted lines, we see no benefit of LMDz6A vs. LMDz5A, since version 6A increases the absolute bias of SURF for eight programs (three in Brazil – RBA-B, ALF, and TAB –, CALNEX2010, DISCOVER-AQ, ACT, THD and LEF) and improves it for four of them (the fourth Brazilian site – SAN –, SEAC4RS, KORUS-AQ, ETL). There is no obvious

consistency between the changes brought by LMDz6A to the surface-based inversion and those brought to the GOSAT-based inversion. For SAN and SENEX2013, the two surface-based inversions have larger absolute biases than the GOSAT-based ones, but perform better for 11 other sites. The poor man's inversion shows the worse biases north of 45°N, but usually performs better than the GOSAT-based inversion in the Southern Hemisphere, likely helped by the tuning with the South Pole Observatory data. Between the Equator and 45°N, the relative performance of the poor man's inversion is uneven but it is

usually not as good as SURF. In terms of standard deviation (bottom row), the surface-based inversions have the smallest ones.

There are 26 aircraft programs in the OCO-2 period. They challenge SURF a bit less (Figure 8) than for the GOSAT period: apart from INPE, GSFC and KORUS-AQ (12% of the programs), all absolute SURF biases are less than 0.45 ppm, while seven programs (19% of the programs, i.e. SAN, SENEX2013, KORUS-AQ, DISCOVER-AQ, HIL, AAO, and CAR) exceeded this threshold previously. The relatively close flux estimates between SURF and OCO-2 inversions (Figure 2 - Figure

*6*) translate into relatively close performance compared to the aircraft. SURF performs better than OCO-2 for INPE and ACT in terms of biases, and worse for GSFC and KORUS-AQ. The poor man's simulation has lesser skill than in the GOSAT period:,it performs much worse than the surface-based and the OCO-2-based inversions in the Northern Hemisphere, and comparably or better in the Southern one. If we combine all measurements together, the root-mean-square difference for the OCO-2-based and the surface-based inversions varies only between 1.51 and 1.56 ppm. The standard deviations are

comparable between the OCO-2-based inversions and the surface-based inversions. LMDz6A improves the SURF biases for KORUS-AQ and degrades them at three other ones (INPE, LEF and ETL). This lack of improvement also appears for OCO-2 (degradation at INPE, KORUS-AQ, and ABOVE). The statistics for four programs (ORCAS, KORUS-AQ, ACT and SONGNEX2015) are directly comparable between the two periods because the corresponding data are fully in both of them: in all four, OCO-2 performs better than GOSAT.

Figure 9 reformulates the bias statistics of Figure 8 on a map of the differences between the absolute biases of inversions OCO-2 and SURF. Like for the program biases, some points are more robust than others (due to varying amount of data), but there is some large-scale coherence, with better performance of SURF in the Southern Hemisphere (as could already be seen in Figure 7 and Figure 8) and in central and Eastern US, while OCO-2 yields smaller biases in the Northern Hemisphere sub-tropics and in Europe. Other parts of the globe are less consistent such as the western Pacific edge or boreal America.

### 3.4. Pixel attribution

Liu and Bowman (2016) proposed a method to quantify the impact of flux changes over the globe on the corresponding change in the mean squared error (MSE) of the transport model simulation with respect to $n$ independent measurements. They demonstrated it in the case of the flux changes from their prior values to their posterior values within the approximations of a linear transport model $\mathbf{M}$ (including the sampling operator at measurement time and location) and of an unchanged initial state

of $CO_2$. It is actually valid for other types of changes within an inversion, provided they respect the tangent-linear hypothesis for the transport model. The change in the MSE ($\delta MSE$) is expressed as a finite sum of terms. There is one term for each element $i$ of the inversion control vector (i.e., a $CO_2$ flux at a given time and location, or some part of the 3D initial state of $CO_2$). Term $i$ is the product of the corresponding change in the control vector (i.e. a scalar $\delta f_i$), times the corresponding row of the transpose of the linear model $\mathbf{M}$, times (dot product here) the vector of the sum of the differences between the two model

simulations (one, $\mathbf{C_1}$, before the change in the control vector and one, $\mathbf{C_2}$, after the change) and all verification measurements ($\delta\mathbf{C_1}+\delta\mathbf{C_2}$, both vectors with dimension $n$):





$$\delta MSE = \Sigma_i \, \delta f_i \, [ \, \mathbf{M}^T \, (\delta\mathbf{C_1} + \delta\mathbf{C_2}) \, ]_i \qquad\qquad (1)$$

Interestingly, the second product in this formula can be calculated by the adjoint code of the transport model, if it exists, which is the case for LMDz (Section 2.2). Further detail is given in Liu and Bowman (2016).

We apply this approach to interpret the difference between the OCO-2-based and the surface-based inversions using LMDz5A.
The overall MSE is very similar between both ($1.5^2$ ppm$^2$), but the relative performance still varies in space and time (Figure 8 and Figure 9) and we hope to extract some further insight into the relative merits of each dataset. In practice, we compute $\delta\mathbf{C_1} + \delta\mathbf{C_2}$ using the LMDz model linearized around the inversion prior simulation in order to respect the underlying hypothesis. However, some inconsistencies for the initial state of $CO_2$ could not be completely removed between $\delta\mathbf{C_1}$ and $\delta\mathbf{C_2}$ due to the different starting date of each inversion. The map of the sum of all contributions of the flux changes $\delta f_i$ (from the surface-based
inversion to the OCO-2-based inversion) at a given pixel to the change in MSE ($\delta MSE$) is presented in Figure 10. Positive values occur when the OCO-2-based fluxes increase the MSE relative to the surface-based fluxes. This happens in the Western contiguous US, the northeastern South America, Western Europe, Turkey, the West Siberian Plain and eastern Siberia. Contributions to reduce the MSE (negative values) are mostly in Alaska and the eastern contiguous US, western South America, southern Africa, South and South-east Asia, and Indonesia. No noticeable contribution is seen over the ocean, where
OCO-2 retrievals have not been assimilated. By construction, regions that are not well observed downstream by aircraft have lesser contributions, like in Africa. This feature makes the relative magnitude of the patterns among each other not much informative about the flux quality. We will therefore pay more attention to the sign of the dominant patterns.

The map of Figure 9, which refers to differences in absolute biases within moving windows, is in principle not directly comparable with Figure 10, which refers to MSEs. However, bias changes are much larger than standard deviation changes
(Figure 8) which makes the map of root-mean-square errors (RMSEs, not shown) very similar to Figure 9. Differences between the patterns of Figure 9 in the space of free-tropospheric mole fractions and those of Figure 10 in the space of fluxes are linked to the way $CO_2$ is transported between the surface and the free troposphere. Dominating westerlies outside the Tropics bring the positive flux contributions of Figure 10 to the west of the positive RMSE variations of Figure 9, like from the western to eastern US, or, at a much larger scale, from Eurasia to Alaska. Similarly, negative flux contributions from the eastern US
induce negative RMSE variations in the central North Atlantic Ocean, and tropical easterlies link the negative flux contributions from Southern Africa to the negative RMSE variations in the tropical Atlantic Ocean. The distance between flux signal and free tropospheric signal implies an important role for the transport model in attributing the latter to the former so that these patterns should be considered with caution, as has been the case for inversion systems in general.

## 4. Discussion

Interest in atmospheric $CO_2$ observations has grown dramatically over the last decade, with the hope that they can reliably quantify the evolution of the $CO_2$ sources and sinks. However, a suite of physical and statistical models is needed to estimate the latter from the former. For instance, the link between some of these observations, like the satellite retrievals, and measurement standards is not direct and needs to be empirically made. We also lack measurements dedicated to the development and validation of atmospheric transport models, in particular for subgrid-scale processes. Therefore, the various
underlying models are still in development and our current source-sink estimation capability is not clear: there is no consensus about the latitudinal distribution of the natural carbon fluxes (Le Quéré et al., 2018) or about the carbon budget of relatively well-documented regions like Europe (Reuter et al., 2017). We have defined here quality measures for global inversion systems in order to evaluate the current skill of global inversions, through the example of the CAMS inversion system. By focussing on a specific inversion system, we have avoided the problem of heterogeneity of TransCom-type ensembles, that gather
systems with various degrees of sophistication (resolution of the transport model, size of the control vector), but we still varied the assimilated data (surface or satellite) and the transport model in order to generate a small inversion ensemble.

In practice, quality measures for a data assimilation system must rely on unbiased and independent data (Talagrand, 2015). The property of unbiasedness means that the errors are null on statistical average. The property of independence means that the errors affecting the verification data must not be correlated with the errors affecting the observations that have been used
in the inversion. Ideally, the verification data should be the carbon fluxes to be evaluated, but in the specific case of global inverse systems, the spatial resolution of existing flux observations (of the order of a hundred meters) is much smaller than the spatial resolution of global transport models (larger than a degree). Therefore, one has no option but to evaluate the analysed $CO_2$ fields (that are the combination of the analysed surface fluxes, of an analysed initial state of $CO_2$ and of the transport model used in the inversion) rather than the analysed surface fluxes alone, both of them being related through mass-conserving
transport in the global atmosphere. This can be done with atmospheric observations like those listed in the Introduction: surface



measurements, aircraft measurements, TCCON retrievals, AirCore measurements or satellite retrievals. We remove TCCON and the satellite data from the list on the criterion of unbiasedness (Figure 9 suggests that we are interested here in signals that are smaller than the TCCON trueness), the surface data on the criterion of independence (the surface data in ObsPack-type databases that are well simulated by the transport models are usually assimilated), and the AirCore data because of limited

time and space coverage so far. This leaves aircraft data as an obvious choice to define objective measures of quality of the inversion systems, when they are not assimilated. They have served this role in the past to some extent, starting from Peylin et al. (2006) or Stephens et al. (2007) (see also Pickett-Heaps et al., 2011; Basu et al., 2014; Houweling et al., 2015; Frankenberg et al., 2016; Le Quéré et al., 2018; Crowell et al., 2019), but few aircraft measurement programs have been used so far and, as a consequence, their use has rarely been formulated in terms of quality assurance or quality control processes for atmospheric

inversions. Compared to previous studies, we benefit from a much larger number of aircraft measurements over the globe in the free troposphere (600,000 for the OCO-2 period, twice as much for the GOSAT period) and from more recent satellite retrievals.

We have used data between 2 and 7 km above sea level only, where the age of air varies significantly (Krol et al., 2018). Aircraft data in this region of the atmosphere only sample a portion of the carbon cycle. With their sparse coverage at places,

they may miss some of the tropical flux signal that can reach higher levels within a few days, but flux errors compensate at the global scale such that errors in the Tropics that would not be directly seen will likely induce errors elsewhere that can be seen. Conversely, our 5-km wide layer still represents a large portion of the column observed by the satellites. However, with the use of individual pointwise measurements (rather than profile averages), we hope to have minimized the possible advantage given to the satellite inversions with respect to the surface-based inversions. The gradient between mole fractions in the

boundary layer and the free troposphere is also informative (Stephens et al., 2007). It provides complementary information about inversion quality, provided that the minority of measurements above urban areas or in the vicinity of assimilated surface sites are excluded. This has not been explored here.

For our ensemble of six Bayesian inversion results, we have seen that large differences in the estimated annual subcontinental fluxes (GOSAT-based vs. surface-based results) are paralleled by different quality of fit to the aircraft data, with GOSAT-

based results performing less well. An additional poor man's inversion that simply adjusts very recent bottom-up flux estimates with the annual global growth rate, has larger differences than the surface-based and the OCO-2-based inversions in terms of flux and the aircraft data. Changing the transport model affected the flux estimation only at the scale of TransCom-type regions: no benefit could be seen with respect to aircraft data, despite six years of model development within the CMIP framework by the LMDz team and despite improved nudging meteorological variables between the two versions (from ERA-Interim to ERA-

5). This suggests that LMDz transport errors play a much smaller role in the quality of our inversion results than the choice of assimilated data. In comparison to the GOSAT results, or to previous OCO-2 inversion results (Crowell et al., 2019), OCO-2-based annual fluxes are surprisingly close to the surface-based fluxes (usually within 1 σ of the Bayesian uncertainty of the surface-based fluxes). Consequently, the aircraft data used here do not allow us to distinguish between the quality of OCO-2-based fluxes and surface-based fluxes. The poor man's inversion still performs worse despite the contribution of a recent

dynamic global vegetation model simulation, showing that the OCO-2 performance is not trivial. Following Liu and Bowman (2016), we attribute the simulation error changes in the free troposphere for the OCO-2 period to flux differences in specific regions of the globe. We find a rather homogeneous geographical distribution of the flux performance with OCO-2-based fluxes and surface-based fluxes alternating as the best ones over continental land masses. This adjoint analysis also illustrates the large footprint of our aircraft data in the free troposphere in terms of flux information, which prevents using them for the

evaluation of local fluxes, given our choice of altitude range of 2 -7 km above sea level.

## 5. Conclusions

Within the limitations imposed by the use of two different verification periods, bias-corrected OCO-2 retrievals perform better than GOSAT retrievals in our inversion system. Upstream, both inferred flux time series do not overlap with each other at all scales studied here (for instance in the tropical lands) in terms of both the mean and variability. This prevents us from

computing flux anomalies from one vs. the other. Within the study timeframe, it was not possible to test more than a couple of different versions of the GOSAT retrievals or other ways to assimilate the OCO-2 retrievals. Indeed, each one of our six Bayesian inversions represented a large computational effort that lasted between four and six weeks on a parallel cluster. We could therefore not identify the distinctive asset of OCO-2 vs. GOSAT in our system: either the data density, the data precision, the data trueness (linked both to the quality of the physical retrieval scheme and to its empirical bias-correction), or a

combination of these qualities at once. Further, other GOSAT-based inversions could be more competitive if made differently (e.g., with a different bias-correction), while other OCO-2-based inversions (e.g., with a different transport model or with different retrievals), or ours with ACOS v9 retrievals after our study period (e.g., if the empirical bias-correction is less efficient



for later months), could still be found deficient for carbon specialists. As we have shown, aircraft data can help ranking the skill of these alternative inversion configurations between each other and vs. ours (all data used here, apart from the recent INPE data, are publicly available).

This validation strategy assumes that airborne measurement programs are continued while new satellite observations are made, and that these programs fairly sample the diversity of $CO_2$ plumes in the free troposphere. In this respect, the situation is not satisfactory at present in some parts of the world, like Africa. This validation strategy also implies that aircraft data are kept independent from the inversion system, and therefore that observations dedicated to the free troposphere (aircraft or satellite partial column retrievals) are not assimilated. This is usually the case, for instance because of the challenging characterization of model errors in simulating aircraft profiles or because systematic errors for partial column retrievals are too large. Zhang et al. (2014) or Alden et al. (2016) presented a different strategy in which aircraft profile measurements are assimilated: a compromise has to be found between exploiting valuable data directly (in particular in areas void of surface measurements), or keeping them for validation.

Finally, the evidence provided by aircraft measurements in the free troposphere suggests that the quality of OCO-2 retrievals over land is now high enough to provide results that are comparable in credibility to the reference (but sparse) surface air sample network, within the above-discussed limits. For ocean retrievals, this remains unclear as OCO-2 ocean soundings were not tested in this work. The consistency of results from the surface and OCO-2-driven inversions, in stark contrast to the bottom-up fluxes or to the GOSAT-driven inversion, does not seem to be fortuitous. It may reinforce some specific conclusions from the surface network, for instance pertaining to the location of the land sink in latitude during the recent years. Remaining differences between fluxes from these two flux inversion types require further analysis and underline their complementarity. The best results may now be obtained by inversions that simultaneously assimilate both observation types.

**Data availability:**

The aircraft measurements are available from https://www.esrl.noaa.gov/gmd/ccgg/obspack/. The OCO-2 data can be obtained from http://co2.jpl.nasa.gov. They were produced by the OCO-2 project at the Jet Propulsion Laboratory, California Institute of Technology. The CAMS v18r1 product can be obtained from http://atmosphere.copernicus.eu/ and the GOSAT retrievals from http://climate.copernicus.eu/. All inversion results can be obtained on request to copernicus-support@ecmwf.int.

**Author contributions:**

FC designed the experiments and carried them out. FC, MR, CWO, DB and PP analysed the results. MR and AC adapted the LMDz6A model for tracer transport. FC and MR prepared the manuscript with contributions from all co-authors.

**Competing interests:**

The authors declare that they have no conflict of interest.

**Acknowledgements:**

The authors are very grateful to the many people involved in the surface, aircraft and satellite $CO_2$ observations and in the archiving of these data that were kindly made available to them, like H. Bösch and J. Anand at the University of Leicester, the ObsPack team and LaGEE, the Greenhouse Gases Laboratory from INPE. The Carbon in Arctic Reservoirs Vulnerability Experiment (CARVE) is an Earth Ventures (EV-1) investigation, under contract with the National Aeronautics and Space Administration. The Atmospheric Carbon and Transport (ACT) – America project is a NASA Earth Venture Suborbital 2 project funded by NASA's Earth Science Division (Grant NNX15AG76G to Penn State). The authors received funding from the Copernicus Atmosphere Monitoring Service, implemented by the European Centre for Medium-Range Weather Forecasts (ECMWF) on behalf of the European Commission. This work was performed using HPC resources from GENCI-TGCC (Grant 2018-A0050102201). The authors thank H. Bösch, F.-M. Bréon, G. Broquet, J.B. Miller, B. Stephens, J. Peischl and S. Houweling for constructive discussions about these results and how they are presented. B. Baier, K. Davis, L. Gatti, K. McKain, and C.E. Miller provided useful detail about some of the measurement programs.



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



**Table 1 : Aircraft measurement programs used here. Note that programs ALF, PAN, RBA-B, SAN and TEF are gathered under identifier INPE (for *Instituto Nacional de Pesquisas Espaciais*) in Figure 8.**

| Measurement program identifier in ObsPack | Measurement program name | Specific doi | Data providers |
|---|---|---|---|
| AAO | Airborne Aerosol Observatory, Bondville, Illinois (NOAA/ESRL Global Greenhouse Gas Reference Network aircraft program) | | Sweeney, C.; Dlugokencky, E.J. |
| ABOVE | Arctic-Boreal Vulnerability Experiment (NASA Airborne Science) | https://doi.org/10.3334/ORNLDAAC/1658 | Sweeney, C.; McKain, K. |
| ACG | Alaska Coast Guard (NOAA/ESRL Global Greenhouse Gas Reference Network aircraft program) | | Sweeney, C.; McKain, K.; Karion, A.; Dlugokencky, E.J. |
| ACT | Atmospheric Carbon and Transport – America (NASA Airborne Science) | https://doi.org/10.3334/ORNLDAAC/1593 | In situ : Davis, K.J.; DiGangi, J.P.; Yang, M. |
| | https://daac.ornl.gov/cgi-bin/dataset_lister.pl?p=37 | | Flasks: Sweeney, C.; Baier, B.; Lang, P.. |
| ALF | Aircraft Observation of Atmospheric GHG at Alta Floresta, Mato Grosso by LaGEE/INPE | | Gatti, L.V.; Miller, J.B.; Gloor, E.; Peters, W. |
| AOA | Aircraft Observation of Atmospheric trace gases by JMA | | ghg_obs@met.kishou.go.jp |
| BNE | Beaver Crossing, Nebraska (NOAA/ESRL Global Greenhouse Gas Reference Network aircraft program) | | Sweeney, C.; Dlugokencky, E.J. |
| CALNEX2010 | California Nexus 2010 (NASA Airborne Science) | | Ryerson, T.B.; Peischl, J.; Aikin, K.C. |
| CAR | Briggsdale, Colorado (NOAA/ESRL Global Greenhouse Gas Reference Network aircraft program) | | Sweeney, C.; Dlugokencky, E.J. |
| CMA | Offshore Cape May, New Jersey (NOAA/ESRL Global Greenhouse Gas Reference Network aircraft program) | | Sweeney, C.; Dlugokencky, E.J. |
| CON | CONTRAIL (Comprehensive Observation Network for TRace gases by AIrLiner) | http://dx.doi.org/10.17595/20180208.001 | Machida, T.; Matsueda, H.; Sawa, Y. Niwa, Y. |
| CRV | Carbon in Arctic Reservoirs Vulnerability Experiment (CARVE, NASA Airborne Science) | | Sweeney, C.; Karion, A.; Miller, J.B.; Miller, C.E; Dlugokencky, E.J. |



| | | | |
|---|---|---|---|
| DC3 | Deep Convective Clouds and Chemistry (NASA Airborne Science) | | Chen, G.; DiGangi, J.P.; Beyersdorf, A. |
| DISCOVER-AQ | Deriving Information on Surface Conditions from Column and Vertically Resolved Observations Relevant to Air Quality (NASA Airborne Science) | | Chen, G.; DiGangi, J.P.; Yang, M. |
| DND | Dahlen, North Dakota (NOAA/ESRL Global Greenhouse Gas Reference Network aircraft program) | | Sweeney, C.; Dlugokencky, E.J. |
| ESP | Estevan Point, British Columbia (NOAA/ESRL Global Greenhouse Gas Reference Network aircraft program) | | Sweeney, C.; Dlugokencky, E.J. |
| ETL | East Trout Lake, Saskatchewan (NOAA/ESRL Global Greenhouse Gas Reference Network aircraft program) | | Sweeney, C.; Dlugokencky, E.J. |
| GSFC | NASA GODDARD Space Flight Center Aircraft Campaign | | Kawa, S.R.; Abshire, J.B.; Riris, H |
| HIL | Homer, Illinois (NOAA/ESRL Global Greenhouse Gas Reference Network aircraft program) | | Sweeney, C.; Dlugokencky, E.J. |
| HIP | HIPPO (HIAPER Pole-to-Pole Observations) | https://doi.org/10.3334/CDIAC/HIPPO_010 | Wofsy, S.C.; Stephens, B.B.; Elkins, J.W.; Hintsa, E.J.; Moore, F. |
| KORUS-AQ | Korea-United States Air Quality Study (NASA Airborne Science) | | Chen, G.; DiGangi, J.P.; Shook, M. |
| LEF | Park Falls, Wisconsin (NOAA/ESRL Global Greenhouse Gas Reference Network aircraft program) | | Sweeney, C.; Dlugokencky, E.J. |
| NHA | Offshore Portsmouth, New Hampshire (NOAA/ESRL Global Greenhouse Gas Reference Network aircraft program) | | Sweeney, C.; Dlugokencky, E.J. |
| ORC | ORCAS ($O_2$/$N_2$ Ratio and $CO_2$ Airborne Southern Ocean Study) | https://doi.org/10.5065/D6SB445X | Stephens, B.B.; Sweeney, C.; McKain, K.; Kort, E.A. |
| PAN | Aircraft Observation of Atmospheric GHG at Pantanal, Mato grosso do Sul by LaGEE/INPE | | Gatti, L.V.; Miller, J.B.; Gloor, E.; Peters, W. |
| PFA | Poker Flat, Alaska (NOAA/ESRL Global Greenhouse Gas Reference Network aircraft program) | | Sweeney, C.; Dlugokencky, E.J. |
| RBA-B | Aircraft Observation of Atmospheric GHG at Rio Branco, Acre by LaGEE/INPE | | Gatti, L.V.; Miller, J.B.; Gloor, E. ; Peters, W. |



| | | | |
|---|---|---|---|
| RTA | Rarotonga (NOAA/ESRL Global Greenhouse Gas Reference Network aircraft program) | | Sweeney, C.; Dlugokencky, E.J. |
| SAN | Aircraft Observation of Atmospheric GHG at Santarém, Pará by LaGEE/INPE | | Gatti, L.V.; Miller, J.B.; Gloor, E.; Peters, W. |
| SCA | Offshore Charleston, South Carolina (NOAA/ESRL Global Greenhouse Gas Reference Network aircraft program) | | Sweeney, C.; Dlugokencky, E.J. |
| SEAC4RS | Studies of Emissions and Atmospheric Composition, Clouds and Climate Coupling by Regional Surveys (NASA Airborne Science) | | Chen, G.; DiGangi, J.P.; Beyersdorf, A. |
| SENEX2013 | Southeast Nexus 2013 (air campaign) | | Ryerson, T.B.; Peischl, J.; Aikin, K.C. |
| SGP | Southern Great Plains, Oklahoma (NOAA/ESRL Global Greenhouse Gas Reference Network aircraft program) | | Sweeney, C.; Dlugokencky, E.J.; Biraud, S. |
| SONGNEX2015 | Shale Oil and Natural Gas Nexus 2015 (air campaign) | | Ryerson, T.B.; Peischl, J.; Aikin, K.C. |
| TAB | Aircraft Observation of Atmospheric GHG at Tabatinga, Amazonas by LaGEE/INPE | | Gatti, L.V.; Gloor, E.; Miller, J.B.; Peters, W. |
| TEF | Aircraft Observation of Atmospheric GHG at Tefe, Amazonas by LaGEE/INPE | | Gatti, L.V.; Gloor, E.; Miller, J.B.; Peters, W. |
| TGC | Offshore Corpus Christi, Texas (NOAA/ESRL Global Greenhouse Gas Reference Network aircraft program) | | Sweeney, C.; Dlugokencky, E.J. |
| THD | Trinidad Head, California (NOAA/ESRL Global Greenhouse Gas Reference Network aircraft program) | | Sweeney, C.; Dlugokencky, E.J. |
| TOM | ATom, Atmospheric Tomography Mission (NASA Airborne Science) | https://doi.org/10.3334/ORNLDAAC/1593 | McKain, K.; Sweeney, C. |
| WBI | West Branch, Iowa (NOAA/ESRL Global Greenhouse Gas Reference Network aircraft program) | | Sweeney, C.; Dlugokencky, E.J. |





**Figure 1 : Location of the aircraft measurements used in the free troposphere for the two verification periods. Note that the two periods overlap by 22 months, so that many data appear on both maps.**

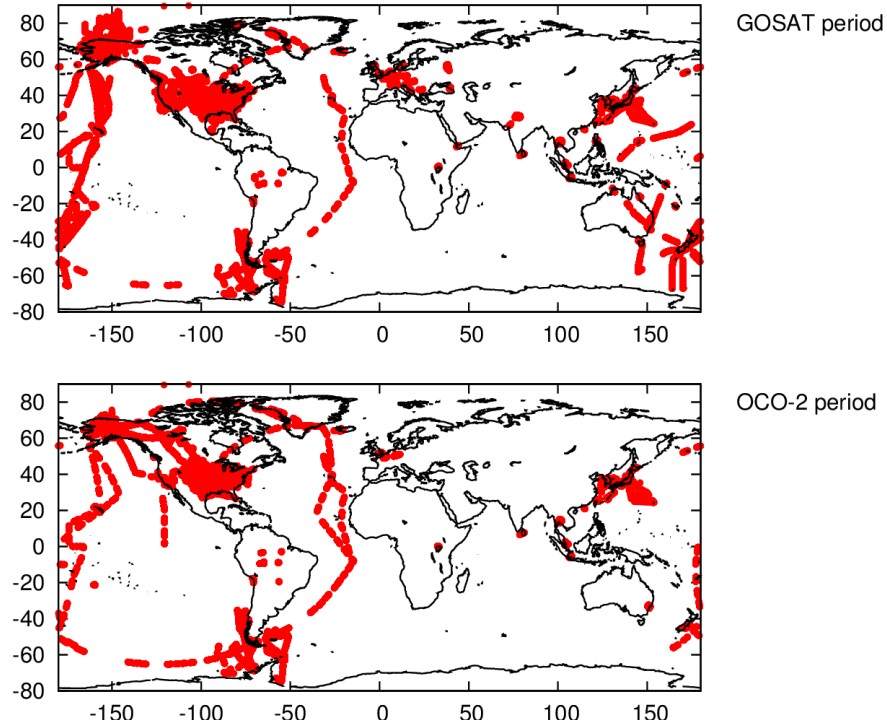



**Figure 2 : Time series of Inferred natural CO₂ annual flux (without the prescribed fossil fuel emissions) between 2004 and 2017, averaged over the globe or over all lands or oceans. In the case of lands and oceans three broad latitude bands are also defined: northern extra-Tropics (north of 25ºN), Tropics (within 25º of the Equator), and southern extra-Tropics (south of 25ºS). Inversions with LMDz5A (LMDz6A) are shown in continuous (dashed) coloured lines. In the**
5 **sign convention, positive fluxes correspond to a net carbon source into the atmosphere. The last year of the GOSAT inversions (2016) is not represented because of likely edge effects. Note that the prior fluxes are zero over land at this temporal scale (see Section 2.2) and that they are equal to curve "Poor Man" over the ocean (see Section 2.5).**

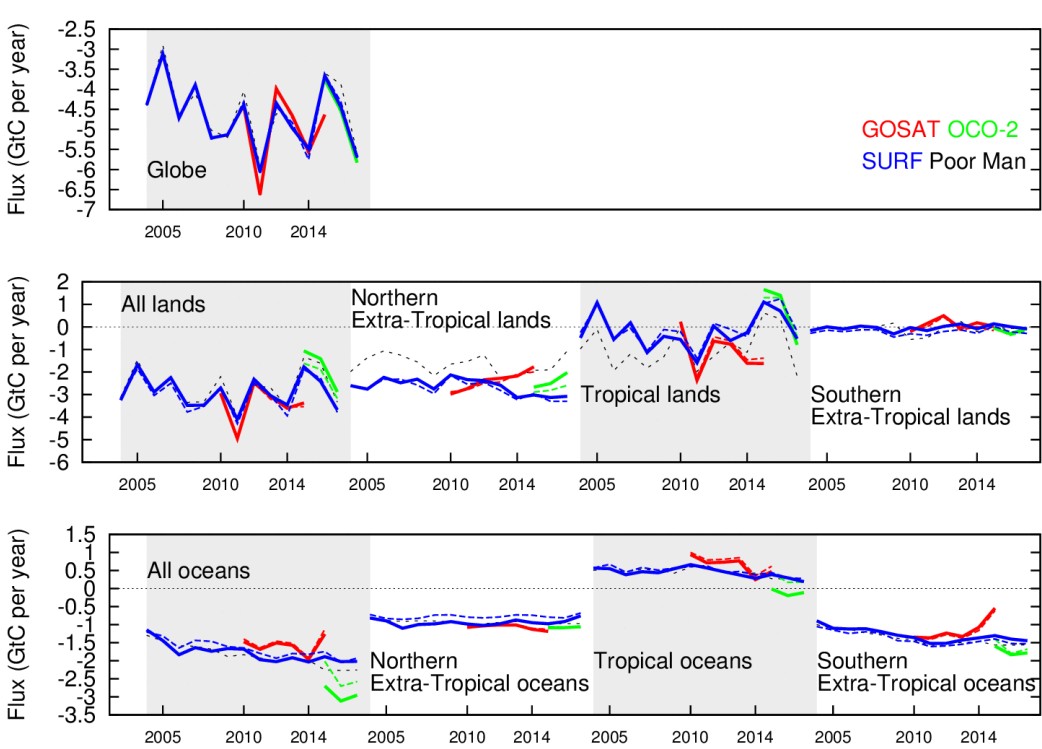



**Figure 3 : Time series of Inferred natural CO₂ annual flux (without the prescribed fossil fuel emissions) between 2004 and 2017, averaged over TransCom 3 land regions. Inversions with LMDz5A (LMDz6A) are shown in continuous (dashed) coloured lines. In the sign convention, positive fluxes correspond to a net carbon source into the atmosphere. The last year of the GOSAT inversions (2016) is not represented because of likely edge effects. Note that the prior fluxes are zero over land at this temporal scale (see Section 2.2).**

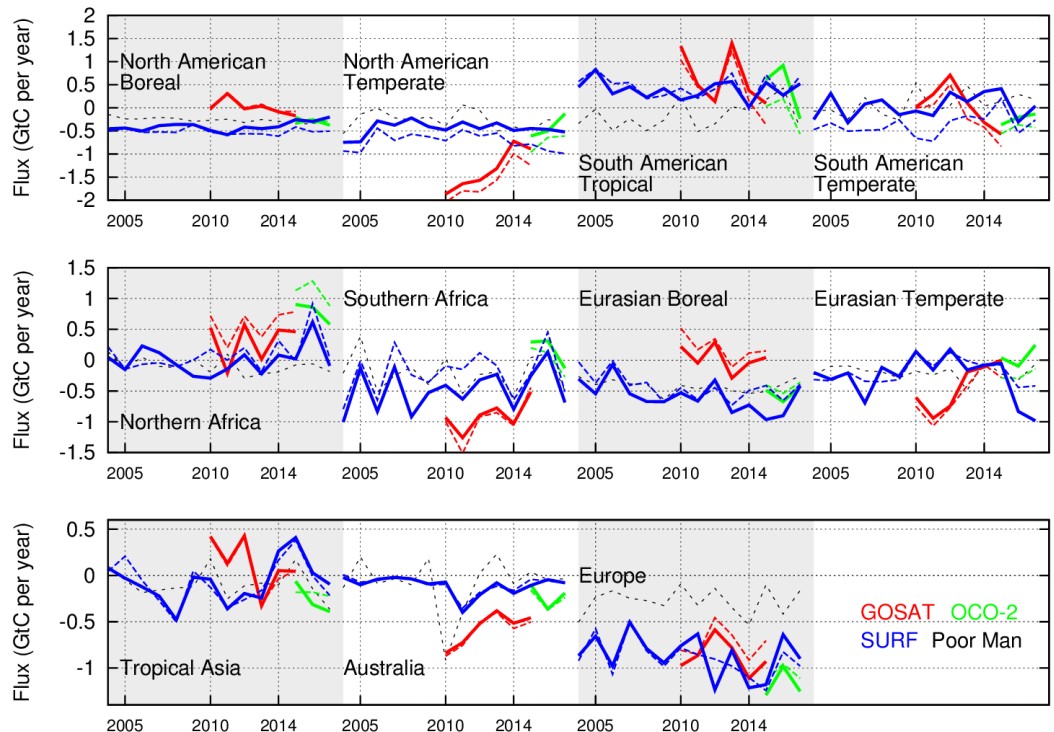


**Figure 4 : Same as Figure 3 but for oceanic regions. Note that the prior fluxes over the ocean are equal to curve "Poor Man" (see Section 2.5).**

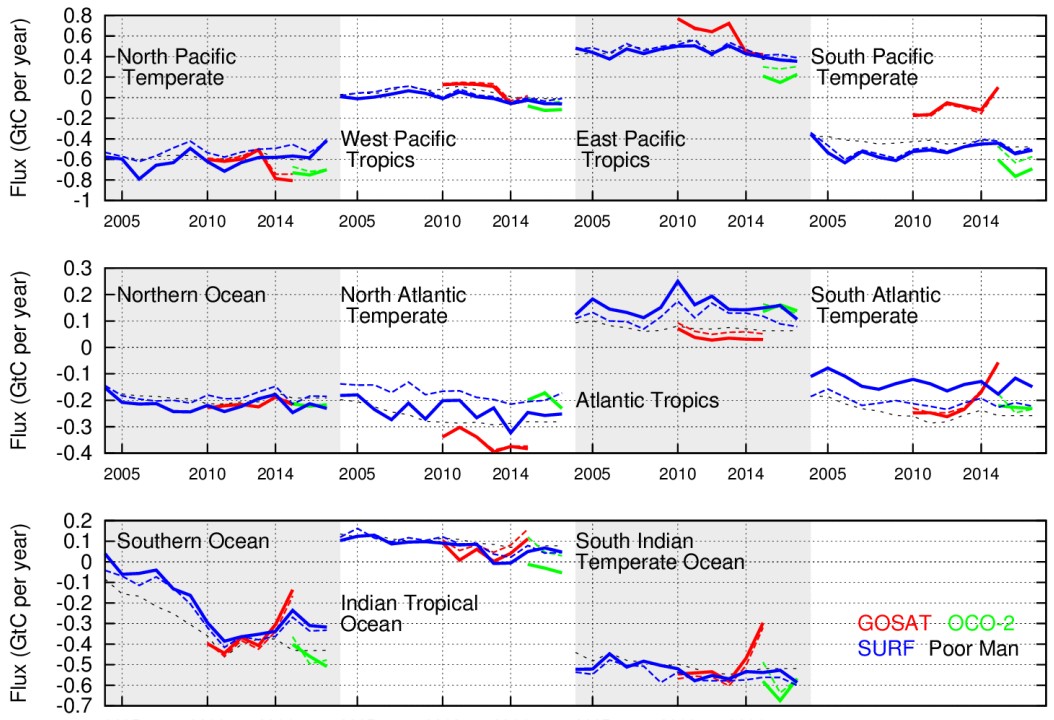





**Figure 5 : Ratio of the absolute difference (δflux) between the OCO-2-based annual fluxes and the surface-based annual fluxes to the Bayesian posterior flux uncertainty for the surface-based fluxes (σᵃ), in %, for years 2015, 2016, and 2017. Both inversions correspond to LMDz6A.**

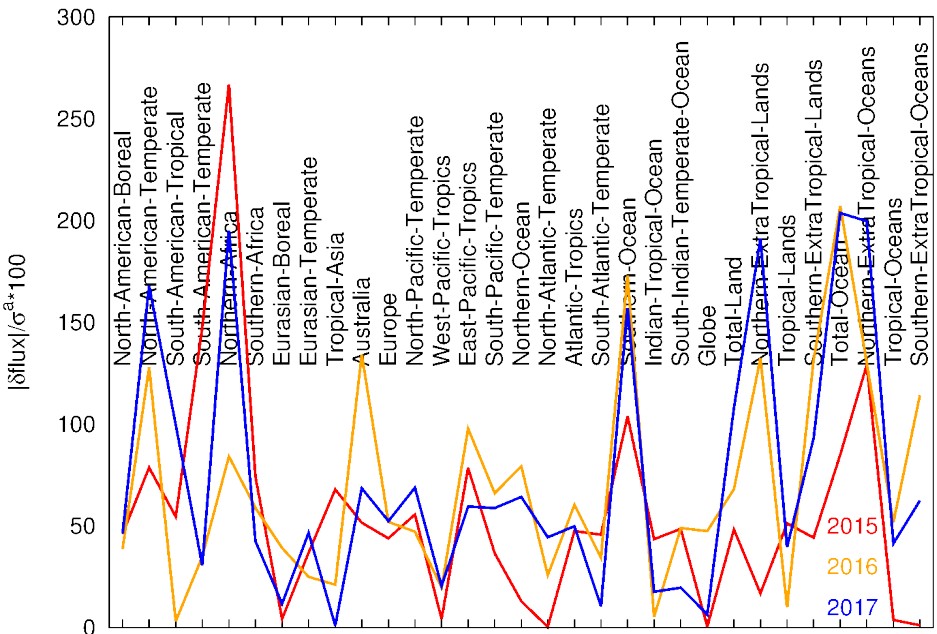





**Figure 6 : Grid-point budget of the natural CO₂ fluxes for the year 2015. In the sign convention, positive fluxes correspond to a net carbon source into the atmosphere.**

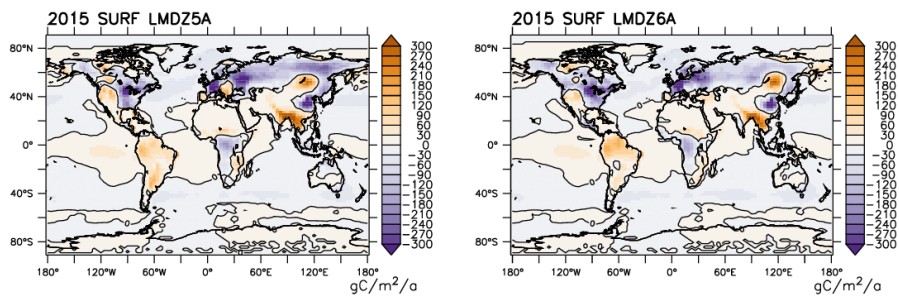

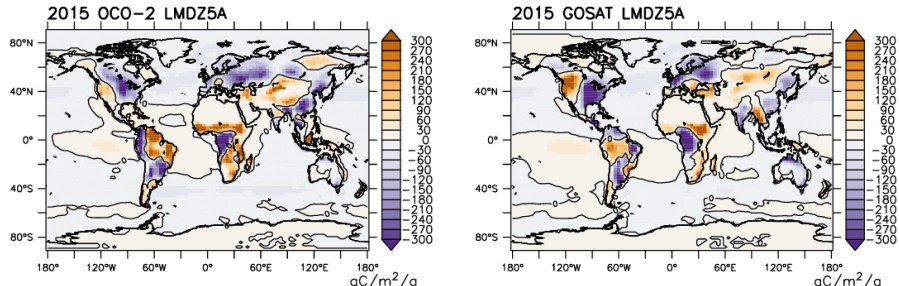



**Figure 7 :** **Model-minus-observation absolute differences and standard deviations over the GOSAT period per measurement program for the surface-based inversion (SURF, red line), the GOSAT-based inversion (GOSAT, blue line) and the poor man's inversion (shaded area). Inversions with LMDz5A (LMDz6A) are shown in continuous (dashed) coloured lines. The number of measurement per site, campaign or program varies between 113 (BNE) and 901,846 (CON). The program definition is given in Table 1. They are ranked by increasing mean latitude (North is on the right), irrespective of their latitudinal coverage (which is large of several tens of degrees for ORC, TOM, HIP and CON). These mean latitudes are shown in the middle of the panel. For each program, a green circle appears in the upper panel if the difference between the GOSAT bias and the SURF bias using LMDz5A is statistically significant (see the main text for a definition) and exceeds 0.15 ppm. Similarly, a blue (red) circle indicates that the difference between LMDz5A and LMDz6A for SURF (GOSAT) is statistically significant and exceeds 0.15 ppm.**

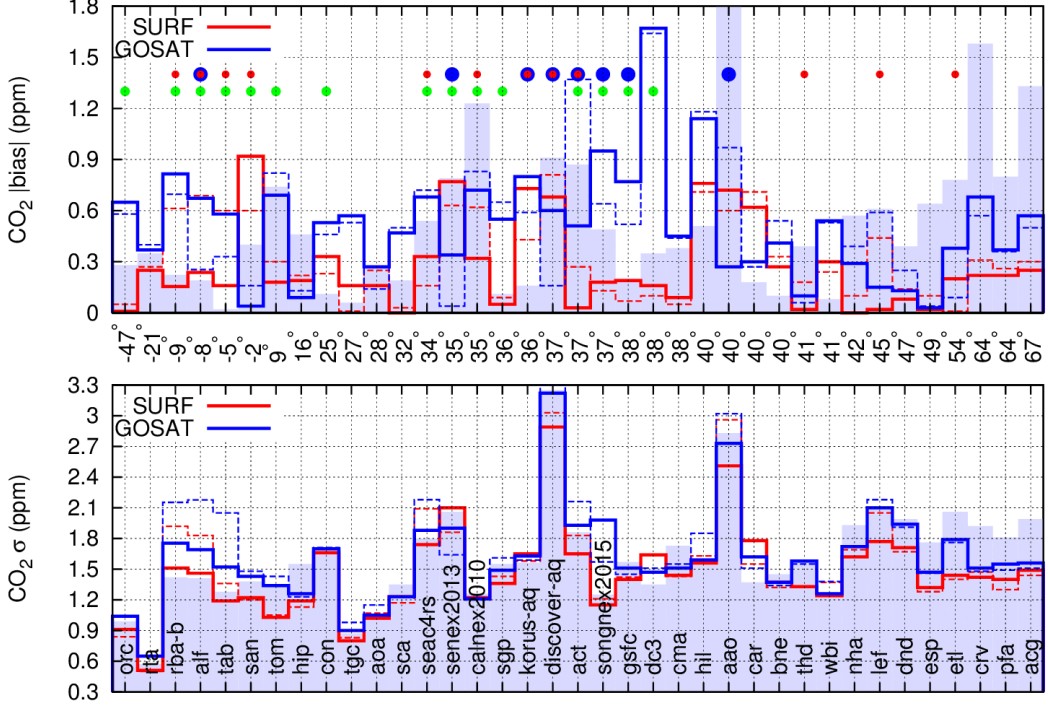



**Figure 8 : Same as Figure 7 for the OCO-2 period. The number of measurements per program varies here between 133 (CRV) and 211,358 (CON).**

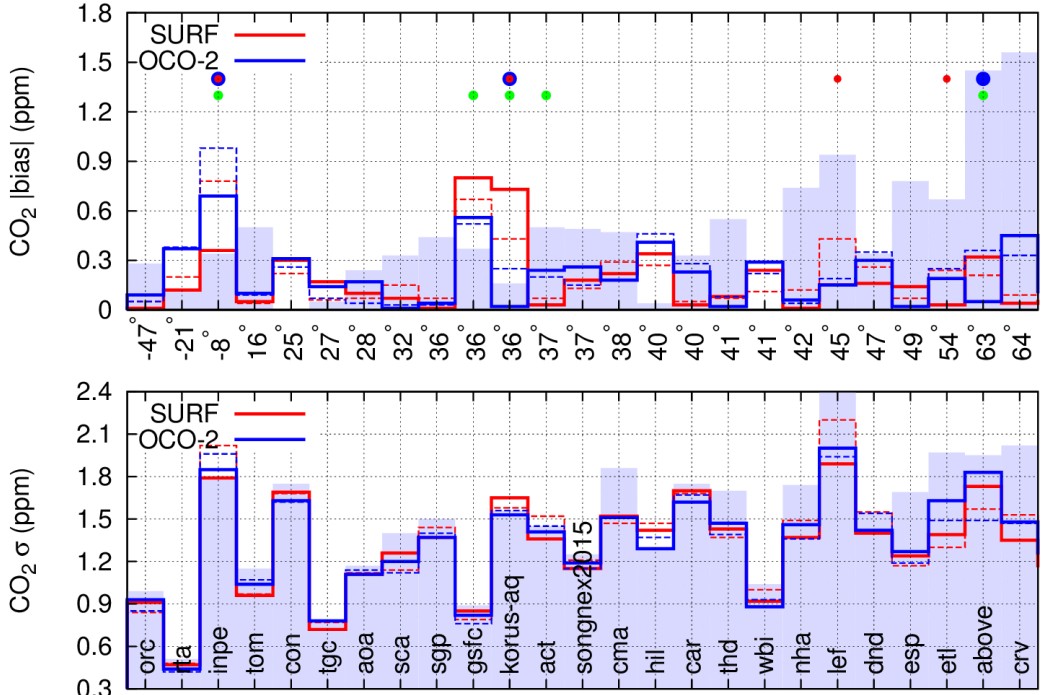



**Figure 9 : Difference between the model-minus-observation absolute differences in 10º moving windows (top). Negative (positive) values denote areas where the OCO-2-based inversion has smaller (larger) biases than the surface-based inversion. Both inversions use LMDz5A. The bottom figure gives the number of data that contribute to the bias computation in each 10º moving window. Biases are computed only in the windows where there are more than 100 measurements.**

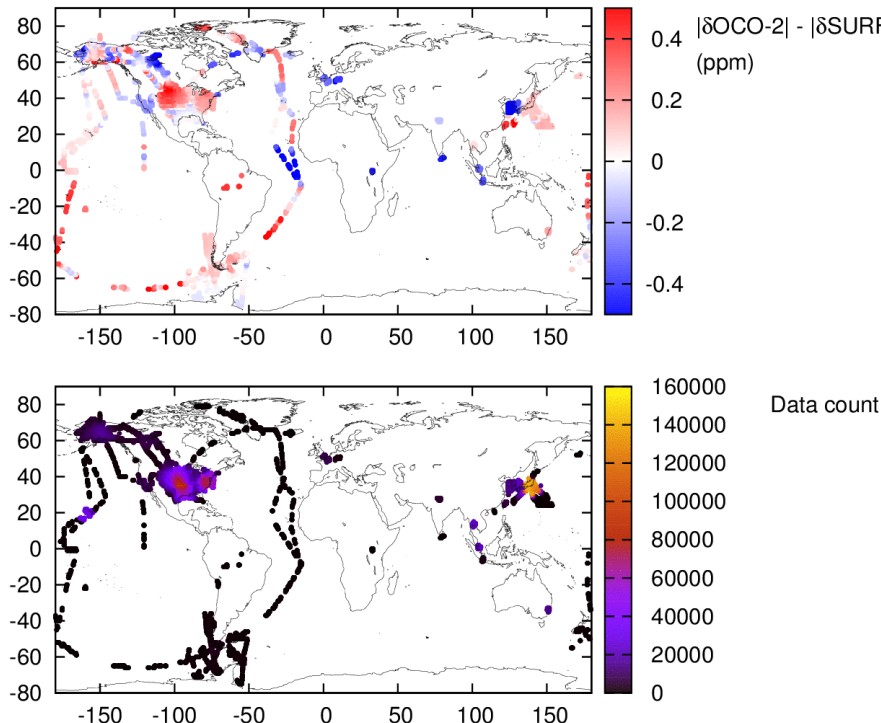





**Figure 10 : Contribution of the grid-point flux changes to the change in the variance of $CO_2$ model-measurement differences between the OCO-2-based inversion and the surface-based inversion (variance of the former minus variance of the latter), in $ppm^2$. Both inversions use LMDz5A. Note that the fluxes themselves are illustrated in the left column of Figure 6.**

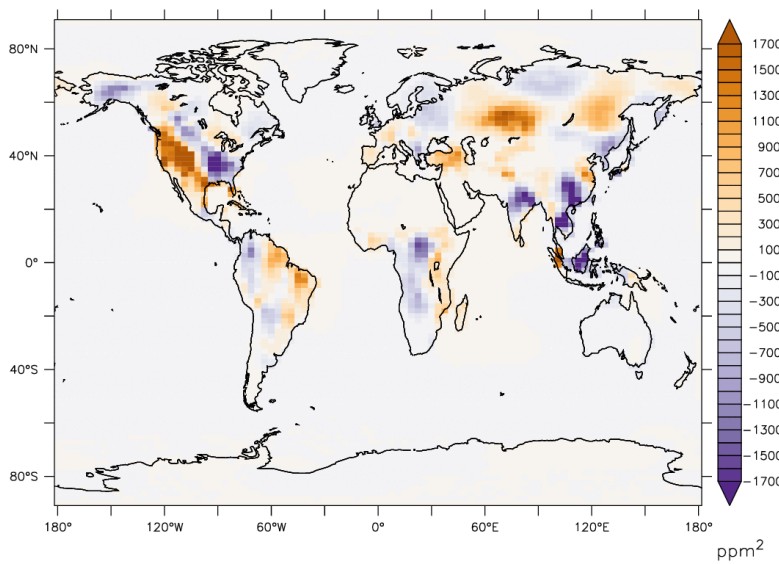

