# Peer review of "Objective evaluation of surface- and satellite-driven CO2 atmospheric inversions"

_Atmospheric Chemistry and Physics, 2019_

## Referee Comment (RC1) · Anonymous Referee #1 · 5 Jun 2019

General comment:

The article presents global inversion results for an ensemble of 6 inversion set-ups, comprising observations from OCO-2, GOSAT and in situ measurements, and using two versions of the transport model. The presented results show that the OCO-2 XCO2 retrievals have matured sufficiently to yield inversion results based on satellite observations that are of similar quality as inversions based on (sparse) in situ measurements. This is a very promising result, paving the way for future satellite inversions. However, the main point the authors want to make consists of "defining quality measures for global inversion systems in order to evaluate the current skill of global inversions". This effort, in my opinion, falls short. Whereas the manuscript presents a nice analysis of an inversion ensemble, a clearly defined set of quality measures is hard to extract

from the text, nor is the main verification approach (comparison against aircraft observations) very novel. It is unclear to this reader what exactly these proposed quality measures look like, or how they should be applied to objectively benchmark the quality of inversion results. Overall this is a nice article, but the focus of it is not very clear: is it the definition of quality measures, or the analysis of a set of global inversions in view of the effect of the transport model and assimilated observations? This ambiguity unfortunately causes neither of these points to come across sufficiently strong.

Specific comments:

- The effect of the transport model on the inversion result is included in the analysis by using two different versions of the same model. The results indicate that this difference has very little effect on the obtained surface fluxes, from which the conclusion seems to be drawn that transport errors have a much lesser influence on the quality of the inversion than do the assimilated observations. Although this seems to be the case for the analyzed inversions, it is well known that in general the transport model is one of the largest sources of difference between inversions (as for example shown again recently in Schuh 2019). This analysis is therefore misleading: the chosen variations in transport model are likely not representative for the actual transport variations typically observed between different inversion set-ups. In order to include these results, they should therefore be situated in the context of an analysis of the decrease in model error between both versions, and preferably also compared against the typical spread between transport models.

- The OCO-2 MIP (Crowell 2019) is touched upon rather briefly, but it would make the analysis stronger if more of the obtained results are put into perspective with respect to the inversion ensemble analyzed in the MIP.

- One aspect which might be given some more attention in the manuscript is the presence of spatial patterns in the optimized surface fluxes due to the used prior, another aspect that typically causes large differences between inversion results. It is mentioned that the OCO-2 and in situ based fluxes look similar, can this be due to the prior pattern being dominant over observation patters in regions with sparse observational coverage?

- The aircraft verification data has especially large coverage over North America. Is the coverage over the other continents sufficient to make statements about the inversion quality? How is this considered when analyzing the root-mean-square difference (Line 28 p7)? Are these values a representative quality measure, or biased towards quality above North America?

- Similar to figure 1, it would be nice to include maps with the observational coverage (and if possible quality) of the different sets of assimilated observations, and to refer to this information in the discussion of the inversion results.

- Line 32, page 4: please include a little more information about the used transport uncertainty.

- On page 4 (lines 37-39) it is mentioned that the bias of OCO-2 ocean observations is still too large for these observations to be included in the inversion, based on initial testing. It would be interesting to include a little more information here: how significant is the bias and the effect of it on optimized fluxes?

- Line 2 on page 5 states that outliers are rejected: are these considered outliers in the aircraft data, or in the simulation result? If the latter, is there a pattern to be discovered? And are the same observations removed for all inversions in the ensemble?

- Line 4 p6: this is not entirely clear: Do you mean that the grid cell value is used as simulated value for the verification data?

- Line 33 p9: if the aircraft data does not allow to distinguish between the quality of OCO-2 and in situ-based fluxes, what does this imply for future quality assessments of inversions? One might expect that the quality of inversion results will increase in the coming years, both through the availability of high-quality satellite data, and through

improvements in inversion set-ups. Would this mean that the presented aircraft verifi-cation approach will no longer be useful to distinguish between their quality?

Small comments / technical corrections:

- Please use ppm/yr instead of ppm/a

- Lines 47-51 p7: it would be clearer if you move these after eq(1), especially since the equation is now on the next page.

- Figure 6 might be clearer if you remove the edges of the contourplot, as they now look messy in combination with the edges of the continents.

- Figures 7&8: please start the y-axis of both plots from zero. It would be insightful to include the 0.15ppm benchmark to show which biases fall within the measurement uncertainty.
* * *

---

## Referee Comment (RC2) · Anonymous Referee #2 · 7 Jun 2019

This paper explores the flux constraint of the in situ network, GOSAT, and OCO-2 independently using two different transport models. As such, this paper is extremely important to the community as a benchmark of progress on satellite remote sensing and the relative information in different sensors on the carbon cycle.

The paper is well-written and thorough, and I recommend publication after a few minor additions and/or corrections.

One issue I can foresee is the use of the OCFP XCO2 product for GOSAT and the OCO-2 FP algorithm for OCO-2. A major conclusion of this paper is that GOSAT has serious issues with regard to its carbon cycle constraint. A paper by Takagi et al, perhaps in 2015, showed large differences between the different retrieval algorithms, and you might actually get similar results between OCO-2 and GOSAT if you were to

use the same retrieval algorithm. I believe this at least warrants some discussion in the manuscript.

Also, the small sensitivity to the two different transport models is a significant finding, as other studies have shown a very large sensitivity to transport. Can you comment on this lack of sensitivity? It is due to similarities between Era-Interim and Era-5? Prior flux constraint? You comment that LMDz5a and LMDz6a are very different models. I realize that this evaluation could be a series of papers on its own.

This reviewer especially appreciates the thorough evaluation by aircraft data, though the lack of TCCON evaluation is surprising. Has this evaluation been done? Despite some dependence between the space-based sensors and TCCON, other works have shown that surface inversions can at times actually agree better with TCCON.

Minor changes:

Page 2, Line 35: Should be Schuh et al, 2019

Page 6, Line 19: Could this be an artifact of the way that the NOAA growth rate is calculated (in the marine bdy layer or at Mauna Loa) vs. a global flux?

---

## Referee Comment (RC3) · Anonymous Referee #3 · 24 Jun 2019

The authors of this paper aim at evaluating whether airborne CO2 measurements made in the free troposphere can distinguish six flux inversion results that are based on two different transport model versions and three independent CO2 datasets. The three CO2 datasets used here are a selection of surface-based CO2 measurements and column-mean CO2 concentrations retrieved from GOSAT and OCO-2 satellite observations. The authors presented the evaluation result in terms of the mean of differences between the airborne measurement and corresponding modeled posterior concentration at each airborne site.

Specific comments:

1. The authors achieved their goal of distinguishing the six independent flux inversion results with the free troposphere airborne CO2 measurements, but they go further to

address the relative merits and demerits of using GOSAT and OCO-2 retrieval datasets based on their evaluation results. The current design/setup of the experiment, however, is too limited to discuss that topic; the two satellite retrieval datasets were evaluated over different time periods and with different amounts of airborne observations at different locations/sites. Indeed, the authors acknowledge in the conclusion chapter that the current experiment can be expanded to 1) cover longer periods in which both satellite retrieval datasets overlap, 2) understand the impact of differences among multiple GOSAT/OCO-2 retrieval algorithms available, 3) test out other approaches to handle the dense OCO-2 retrievals, and 4) assess the impact of OCO-2/GOSAT differences in data density, data precision, and quality in CO2 retrieving and bias correction. Items 2-4 are particularly essential topics that need to be explored. I would encourage the authors to go to that depth, if they are to touch on the merit/demerit topic.

2. I am left with questions regarding gaps found between the time series of annual fluxes by GOSAT and OCO-2 presented on Figures 2 through 4. In 2015, the only year the flux estimates by the two satellites overlap, larger gaps are found in the time series for Northern and Southern Africa, Eurasian Boreal, Australia, and South American Tropical (lager than those of N. American Boreal/Temperate, Eurasian Temperate). These are regions where surface-based CO2 measurement sites are sparse, as indicated in Chevallier 2018, and also where evaluation by the airborne CO2 measurement is limited (Figure 9; most of the airborne evaluation is concentrated over N. America). South American airborne sites are among the very few sites that are found over those regions under-sampled by surface measurement networks. At these site, CO2 biases are shown to be larger both in the OCO-2 and GOSAT cases (>0.6 ppm; INPE (Figure 8) and RBA-B (Figure 7) sites). Do both the OCO-2 and GOSAT CO2 biases have the same signs (+/-) at these South American sites? Can this help explain the gaps in the regional flux time series? Do the flux gaps fall within the range of flux uncertainties? What are other reasons that may explain these gaps? What Figures 2 through 4 show poses readers a question of whether column-mean CO2 retrievals from multiple satellite missions can be mixed in CO2 source-sink studies. I think these are worth

discussing in depth in this paper.

Minor comments:

1. Figure 2 caption: Please explain in the main text what "edge effects" are.

2. Figure 7: The SURF values are shown in red color in the bar chart, but blue color is used for SURF circles indicating >0.15 ppm differences. Should this be in red??

---

## Author Comment (AC1) · 9 Jul 2019

We thank the three reviewers for their constructive evaluation of our paper that helps us clarify it. Our comments are inserted within the full review texts hereafter where appropriate. We have marked the corresponding changes to our initial text in green.

**1  Response to Reviewer #1**

General comment:

The article presents global inversion results for an ensemble of 6 inversion set-ups, comprising observations from OCO-2, GOSAT and in situ measurements, and using two versions of the transport model. The presented results show that the OCO-2 XCO2 retrievals have matured sufficiently to yield inversion results based on satellite observations that are of similar quality as inversions based on (sparse) in situ measurements. This is a very promising result, paving the way for future satellite inversions. However, the main point the authors want to make consists of "defining quality measures for global inversion systems in order to evaluate the current skill of global inversions". This effort, in my opinion, falls short. Whereas the manuscript presents a nice analysis of an inversion ensemble, a clearly defined set of quality measures is hard to extract from the text, nor is the main verification approach (comparison against aircraft observations) very novel. It is unclear to this reader what exactly these proposed quality measures look like, or how they should be applied to objectively benchmark the quality of inversion results. Overall this is a nice article, but the focus of it is not very clear: is it the definition of quality measures, or the analysis of a set of global inversions in view of the effect of the transport model and assimilated observations? This ambiguity unfortunately causes neither of these points to come across sufficiently strong.

Our focus is on the definition of quality measures, and we apply it for the analysis of a set of global inversions in view of the effect of the transport model and assimilated observations. We discuss our contribution to our general focus in section Discussion and recall previous uses of aircraft measurements there. We explain that an "objective evaluation" (to use the terms of our title) of global $CO_2$ atmospheric inversions currently has to rely on aircraft measurements, however scarce they are, but that they have hardly served this purpose so far. We hope that our text can stimulate progress in this direction because we believe that objective evaluation is critical for the credibility of atmospheric inversions.

In comparison, the results of our application to the progress of OCO-2-driven inversions

may come stronger but our paper reflects the structure of our scientific approach here (addressing a methodological question with an application). We note that the reviewer does not challenge any specific part of our logic.

Specific comments:

- The effect of the transport model on the inversion result is included in the analysis by using two different versions of the same model. The results indicate that this difference has very little effect on the obtained surface fluxes, from which the conclusion seems to be drawn that transport errors have a much lesser influence on the quality of the inversion than do the assimilated observations. Although this seems to be the case for the analyzed inversions, it is well known that in general the transport model is one of the largest sources of difference between inversions (as for example shown again recently in Schuh 2019). This analysis is therefore misleading: the chosen variations in transport model are likely not representative for the actual transport variations typically observed between different inversion set-ups. In order to include these results, they should therefore be situated in the context of an analysis of the decrease in model error between both versions, and preferably also compared against the typical spread between transport models.

The relatively small impact of transport uncertainty has been expected in the case of the assimilation of OCO-2 after Basu et al. (2018). At first glance, this may be more a surprise for the surface-driven inversions (ibid.). However, Schuh et al. (2019, their Section 4.1), after Yu et al. (2018), highlighted technical artefacts (i.e. regridding problems) to explain some of the large model-to-model differences. LMDz and his parent model are run at the same resolution, so that our model does not suffer for this issue. We also note that only five years ago, the refinement of the vertical grid in our version of LMDz had a major impact (Chevallier et al. 2014). The version with coarse vertical

resolution ran much faster than the new one: it is likely that nobody would have complained if we still used it for model intercomparisons (thereby contributing to enlarge result spread) or other studies, but we have decided to phase it out. Further refinements in the process modelling and in the driving winds have a much more modest effect for the quantities that we study in the paper (inferred annual Transcom-region-scale fluxes). This is consistent with what we saw with the parent model of our off-line model (Remaud et al., 2018, a study that also included a test of the refinement of the 3D model resolution).

Initial text: This result is consistent with our study of the parent model of our off-line model in forward mode (Remaud et al., 2018) and suggests that LMDz transport errors play a much smaller role in the quality of our inversion results than the choice of assimilated data.

Modified text: This result is consistent with our study of the parent model of our off-line model in forward mode (Remaud et al., 2018) and suggests that LMDz transport errors play a much smaller role in the quality of our inversion results than the choice of assimilated data. This may be different for previous versions of LMDz (Chevallier et al., 2014) or some other off-line models (Schuh et al., 2019).

> - The OCO-2 MIP (Crowell 2019) is touched upon rather briefly, but it would make the analysis stronger if more of the obtained results are put into perspective with respect to the inversion ensemble analyzed in the MIP.

Even in the case of our own LMDz-based results, important changes have been made between this contribution to Crowell et al. and the OCO-2 inversions presented here: the retrievals are different both in terms of quality and coverage (O'Dell et al., 2018; see also the analysis of Miller and Michalak, 2019, https://doi.org/10.5194/acp-2019-382, of the difference between the two retrieval products), and the transport model has been upgraded from version 3 to versions 5A and 6A. The most striking difference with respect to the OCO-2 MIP conclusions is the proximity of the OCO-2-based annual

fluxes with the surface-based fluxes, as noted in our discussion section. A finer analysis of the changes would have required running a series of sensitivity tests, but the running time of the inversion system hampers such a purpose, as explained in our conclusion.

> - One aspect which might be given some more attention in the manuscript is the presence of spatial patterns in the optimized surface fluxes due to the used prior, another aspect that typically causes large differences between inversion results. It is mentioned that the OCO-2 and in situ based fluxes look similar, can this be due to the prior pattern being dominant over observation patters in regions with sparse observational coverage?

The paper only presents annual natural fluxes for which our prior fields are uniformly zero over land. All our inversions deviate much from this prior value. This was already noted in Section 3.2 that stated that the OCO-2 inversions were "clearly distinct from the inversion prior fluxes (that are zero on annual average over land), and from the GOSAT-based fluxes". The last words of the sentence, about GOSAT, indicate that the inversion system does not automatically converge to a unique point. Posterior natural fluxes are also influenced by the prior fossil fuel maps, but the spatial patterns of Fig. 6, that we discuss at the end of Section 3.2, do not resemble fossil fuel patterns.

> - The aircraft verification data has especially large coverage over North America. Is the coverage over the other continents sufficient to make statements about the inversion quality?

Most of the aircraft measurement programs are over North America, but the majority of measurements are provided by the CONTRAIL program, as noted in the legends of Figs. 7 ad 8. CONTRAIL samples air at our study altitudes above many cities outside North America. CONTRAIL represents 74% of all data for the GOSAT period and 39% for the OCO-2 period. Still, as noted in the conclusion, "the situation is not satisfactory at present in some parts of the world, like Africa".

How is this considered when analyzing the root-mean-square difference (Line28 p7)? Are these values a representative quality measure, or biased towards quality above North America?

Line28 p7 says "If we combine all measurements together, the root-mean-square difference for the OCO-2-based and the surface-based inversions varies only between 1.51 and 1.56 ppm." If we only take CONTRAIL data, the root-mean-square difference for the OCO-2-based and the surface-based inversions still only varies between 1.60 and 1.67 ppm, which tends to indicate that our conclusions are not biased towards features that are specific to North America.

We have added: "Note that 39% of these data are from CONTRAIL, a program that spreads over all continents."

- Similar to figure 1, it would be nice to include maps with the observational coverage (and if possible quality) of the different sets of assimilated observations, and to refer to this information in the discussion of the inversion results.

Maps of data coverage in our case follow common patterns.

In Section 2.3. "Assimilated observations", we now refer to the maps of Chevallier (2018a) for SURF, to those of Bösch and Anand (2017) for GOSAT and to those of O'Dell et al. (2018) for OCO-2.

- Line 32, page 4: please include a little more information about the used transport uncertainty.

We have added "based on the variability across several models at the OCO-2 sounding locations" before the reference to Crowell et al. (2019).

- On page 4 (lines 37-39) it is mentioned that the bias of OCO-2 ocean observations is still too large for these observations to be included in the inversion, based on initial testing. It would be interesting to include a little more information here: how significant is the bias and the effect of it on optimized fluxes?

We have added the following information: "(annual global ocean sinks about 5 GtC/a, to be compared with the much smaller state-of-the-art estimates in Le Quéré et al., 2018)". In order to be more exhaustive, we have also added "(as are retrievals over inland water or over mixed land-water surfaces)" at the end of the sentence.

- Line 2 on page 5 states that outliers are rejected: are these considered outliers in the aircraft data, or in the simulation result? If the latter, is there a pattern to be discovered? And are the same observations removed for all inversions in the ensemble?

We consider that these measurements sampled local plumes that cannot be represented by our global model. Nearly the same measurements of that sort are identified when compared with each inversion posterior simulation.

We have added "they likely represent very local pollution plumes".

- Line 4 p6: this is not entirely clear: Do you mean that the grid cell value is used as simulated value for the verification data?

Yes.

We have added the reviewer's expression.

- Line 33 p9: if the aircraft data does not allow to distinguish between the quality of OCO-2 and in situ-based fluxes, what does this imply for future

quality assessments of inversions? One might expect that the quality of inversion results will increase in the coming years, both through the availability of high-quality satellite data, and through mprovements in inversion set-ups. Would this mean that the presented aircraft verification approach will no longer be useful to distinguish between their quality?

Consistent with our title, we focus on objective criterions to evaluate global $CO_2$ atmospheric inversions. In our discussion, we explain that they have to rely on aircraft measurements, however scarce they are. To better serve this purpose, in particular in view of future improvements of the inversion systems, we obviously need more aircraft measurements. The analysis could also be extended to aircore measurements whose number over the globe is growing.

We have added "We also need better coverage to accompany the better quality of inversion results expected in the coming years.".

    - Please use ppm/yr instead of ppm/a

We follow ISO 80000-3:2006, that defines a as the symbol for year.

    - Lines 47-51 p7: it would be clearer if you move these after eq (1), especially since the equation is now on the next page.

Done.

    - Figure 6 might be clearer if you remove the edges of the contour plot, as they now look messy in combination with the edges of the continents.

On the other hand, the contour lines help visualizing the flux patterns and are particularly useful over the ocean to spot the outgasing areas. We have left them.

- Figures 78: please start the y-axis of both plots from zero. It would be in-sightful to include the 0.15 ppm benchmark to show which biases fall within the measurement uncertainty.

We agree that such changes would be interesting, but they would also reduce the readability of the existing information, which is already limited. We prefer to keep the figures as they are.

**2 Response to Reviewer #2**

This paper explores the flux constraint of the in situ network, GOSAT, and OCO-2 independently using two different transport models. As such, this paper is extremely important to the community as a benchmark of progress on satellite remote sensing and the relative information in different sensors on the carbon cycle.

The paper is well-written and thorough, and I recommend publication after a few minor additions and/or corrections.

One issue I can foresee is the use of the OCFP XCO2 product for GOSAT and the OCO-2 FP algorithm for OCO-2. A major conclusion of this paper is that GOSAT has serious issues with regard to its carbon cycle constraint.

We disagree on this last point. We tried to be careful to draw general conclusions neither about OCO-2 nor about GOSAT, because our study is limited to two specific retrieval datasets: "Further, other GOSAT-based inversions could be more competitive if made differently (e.g., with a different bias-correction), while other OCO-2-based inversions (e.g., with a different transport model or with different retrievals), or ours with ACOS v9 retrievals after our study period (e.g., if the empirical bias-correction

is less efficient for later months), could still be found deficient for carbon specialists."
(Section Conclusions).

To further clarify this point, in the abstract of the revised version, we have inserted a
short similar statement: "Without any general claim on the usefulness of all OCO-2
retrieval datasets vs. all GOSAT retrieval datasets...".

In the conclusion, we have added two words (in bold here): "Within the limitations im-
posed by the use of two different verification periods, the **tested** bias-corrected OCO-2
retrievals perform better than the **tested** GOSAT retrievals in our inversion system.".

> A paper by Takagi et al, perhaps in 2015, showed large differences between
> the different retrieval algorithms, and you might actually get similar results
> between OCO-2 and GOSAT if you were to use the same retrieval algo-
> rithm. I believe this at least warrants some discussion in the manuscript.

Our response above also addresses this point.

We have added the reference to Takagi et al. (2014) in the conclusion: "despite large
differences between GOSAT retrieval algorithms, Takagi et al., 2014).

> Also, the small sensitivity to the two different transport models is a signifi-
> cant finding, as other studies have shown a very large sensitivity to trans-
> port. Can you comment on this lack of sensitivity? It is due to similarities
> between Era-Interim and Era-5? Prior flux constraint? You comment that
> LMDz5a and LMDz6a are very different models. I realize that this evalua-
> tion could be a series of papers on its own.

In our discussion section, we already mentioned the fact that TransCom-type ensem-
bles "gather systems with various degrees of sophistication (resolution of the transport
model, size of the control vector)". For instance, Schuh et al. (2019, their Section 4.1)

mostly refer to technical problems in the mass flux processing from the parent model to explain strong differences in interhemispheric mixing, after Yu et al. (2018). Strictly speaking, this is not really transport uncertainty: transport processes are better known than what some off-line models simulate. Our study suggests that transport process uncertainty does not play a large role any more for annual and Transcom-type region scale fluxes, but more investigation would be needed to confirm this explanation.

> This reviewer especially appreciates the thorough evaluation by aircraft data, though the lack of TCCON evaluation is surprising. Has this evaluation been done? Despite some dependence between the space-based sensors and TCCON, other works have shown that surface inversions can at times actually agree better with TCCON.

The dependence of the satellite retrievals on the TCCON data through the bias-correction is a serious limitation for the use of TCCON to discriminate between the skill of the various inversions. Further, as we say in the introduction, the fact that "current TCCON retrievals that serve as the best reference for column retrievals with global coverage, have commensurate [sub-ppm] offset uncertainties (Wunch et al., 2015)" hampers clear conclusions. Indeed, our inversion SURF fits the TCCON data mostly within 1 ppm (see Fig. 8 of Chevallier et al., 2017, https://doi.org/10.1002/2017JD026453).

For better clarity, in the above sentence, we have added "site-specific" before "commensurate offset uncertainties". We also refer in Section Discussion to Chevallier (2018) where this issue is further discussed.

> Minor changes: Page 2, Line 35: Should be Schuh et al, 2019

Done.

> Page 6, Line 19: Could this be an artifact of the way that the NOAA growth

rate is calculated (in the marine bdy layer or at Mauna Loa) vs. a global flux?

This is our interpretation as well, but we have not further investigated this point.

**3 Response to Reviewer #3**

The authors of this paper aim at evaluating whether airborne $CO_2$ measurements made in the free troposphere can distinguish six flux inversion results that are based on two different transport model versions and three independent $CO_2$ datasets. The three $CO_2$ datasets used here are a selection of surface-based $CO_2$ measurements and column-mean $CO_2$ concentrations retrieved from GOSAT and OCO-2 satellite observations. The authors presented the evaluation result in terms of the mean of differences between the airborne measurement and corresponding modeled posterior concentration at each airborne site.

Specific comments:

1. The authors achieved their goal of distinguishing the six independent flux inversion results with the free troposphere airborne $CO_2$ measurements, but they go further to address the relative merits and demerits of using GOSAT and OCO-2 retrieval datasets based on their evaluation results. The current design/setup of the experiment, however, is too limited to discuss that topic; the two satellite retrieval datasets were evaluated over different time periods and with different amounts of airborne observations at different locations/sites. Indeed, the authors acknowledge in the conclusion chapter that the current experiment can be expanded to 1) cover longer periods in which both satellite retrieval datasets overlap, 2) under-
stand the impact of differences among multiple GOSAT/OCO-2 retrieval algorithms available, 3) test out other approaches to handle the dense OCO-2 retrievals, and 4) assess the impact of OCO-2/GOSAT differences in data density, data precision, and quality in CO2 retrieving and bias correction. Items2-4 are particularly essential topics that need to be explored. I would encourage the authors to go to that depth, if they are to touch on the merit/demerit topic.

When writing the paper, we had no intention to address the relative merits and demerits of using GOSAT and OCO-2 retrieval datasets in general. We tried to make it clear that we were studying two specific products in particular: "Further, other GOSAT-based inversions could be more competitive if made differently (e.g., with a different bias-correction), while other OCO-2-based inversions (e.g., with a different transport model or with different retrievals), or ours with ACOS v9 retrievals after our study period (e.g., if the empirical bias-correction is less efficient for later months), could still be found deficient for carbon specialists." (Section Conclusions).

To further clarify this point, in the abstract of the revised version, we have inserted a short similar statement: "Without any general claim on the usefulness of all OCO-2 retrieval datasets vs. all GOSAT retrieval datasets. . .".

In the conclusion, we have added two words (in bold here): "Within the limitations imposed by the use of two different verification periods, the **tested** bias-corrected OCO-2 retrievals perform better than the **tested** GOSAT retrievals in our inversion system.".

2. I am left with questions regarding gaps found between the time series of annual fluxes by GOSAT and OCO-2 presented on Figures 2 through 4. In 2015, the only year the flux estimates by the two satellites overlap, larger gaps are found in the time series for Northern and Southern Africa, Eurasian Boreal, Australia, and South American Tropical (lager than those

of N. American Boreal/Temperate, Eurasian Temperate).These are regions where surface-based CO2 measurement sites are sparse, as indicated in Chevallier 2018, and also where evaluation by the airborne CO2 measurement is limited (Figure 9; most of the airborne evaluation is concentrated over N. America). South American airborne sites are among the very few sites that are found over those regions under-sampled by surface measurement networks. At these site, CO2 biases are shown to be larger both in the OCO-2 and GOSAT cases (>0.6 ppm; INPE (Figure8) and RBA-B (Figure 7) sites). Do both the OCO-2 and GOSAT CO2 biases have the same signs (+/-) at these South American sites?

Yes, they have, but the absolute values vary by twofold when considering the same measurements. For the common year (2015), we have 185 aircraft measurements from INPE between 2 and 7 km above sea level. The mean model-observation bias is 0.4, 0.6, 1.0 and 1.1 ppm, respectively for GOSAT/5A, GOSAT/6A, OCO-2/5A and OCO-2/6A.

Can this help explain the gaps in the regional flux time series?

Considering the same 185 measurements (i.e. much less than what is shown in Fig. 7 and 8), the performance of SURF is in-between GOSAT and OCO-2: 0.8 and 1.0 ppm, respectively for 5A and 6A. These statistics are based on a small sample and provide little help to explain the spread of the time series over a large portion of the South American continent.

Do the flux gaps fall within the range of flux uncertainties?

For Tropical South America, the posterior uncertainty of SURF (1$\sigma$) is 1.3 GtC/a and well covers the spread. This large uncertainty is expected since the reviewer points to

regions that are measurement-sparse at the surface. A more general view on this question is provided by Figure 5 that shows the ratio of the absolute difference between the OCO-2 annual fluxes and the SURF annual fluxes to the SURF posterior uncertainty: "For all regions discussed so far, this difference is usually within the Bayesian uncertainty standard deviation (but reaches up to 2.6 times this quantity in Northern Africa for 2015), which means that the difference between the two flux estimates at this scale is mostly not statistically significant" (Section 3.2).

What are other reasons that may explain these gaps?

We acknowledged the fact that "remaining differences between fluxes from these two flux inversion types require further analysis." (Section 5, Conclusions). As long as they are consistent with the posterior uncertainty estimates, they are not surprising: they reflect the limitations of the observation information content as seen through the inverse system.

What Figures 2 through 4 show poses readers a question of whether column-mean CO2 retrievals from multiple satellite missions can be mixed in CO2 source-sink studies. I think these are worth discussing in depth in this paper.

We argued that the satellite-driven time-series presented here cannot be mixed: "Upstream, both inferred flux time series do not overlap with each other at all scales studied here (for instance in the tropical lands) in terms of both the mean and variability. This prevents us from computing flux anomalies from one vs. the other" (Section 5, Conclusions).

Minor comments: 1. Figure 2 caption: Please explain in the main text what "edge effects" are.

We have replaced "because of likely edge effects" by "because it is less constrained at the end by lack of 2017 retrievals here."

> 2. Figure 7: The SURF values are shown in red color in the bar chart, but blue color isused for SURF circles indicating >0.15 ppm differences. Should this be in red??

We thank the reviewer for having spotted the mistake. We have corrected the legend.

---

## Author Response (AR2)

We thank the two reviewers and the editor for their reading of our revised paper and of our answers. Our comments are inserted hereafter within the editor's review text where appropriate.

**Response to the Editor**

*It would be useful to make use of the detailed discussion provided in the response in the paper's body to provide more information to readers.*

*- Following reviewer 3 first two points authors correctly state in the abstract that the paper doesn´t provide claims on the usefulness of Satellite datasets in the retrieval. The answer to the reviewer is important and may be helpful to have a more through discussion to best reflect the strength of the analysis and avoid misunderstandings to the readers.*

*So, I would suggest to make use of the replies given in the Author's response on the discussion of the benefits of OCO vs GOSAT, the outcomes and the possible additional analysis needed – now in the conclusion – in the discussion section with also few additional details to discuss the coherence of Takagi et al results now added as reference.*

*The conclusions may then reflect the abstract statement.*

15 We have added the following statement in the discussion, using some of the expressions of Reviewer 2 (we suppose that the Editor is referring to our response to Reviewer 2 rather than 3) and of our response:

"From our results, we draw general conclusions neither about OCO-2 retrievals nor about GOSAT retrievals, because our study is limited to two specific retrieval datasets. Takagi et al. (2015) showed large differences between different retrieval algorithms for GOSAT, and it is still possible that we would get similar results
20 between OCO-2 and GOSAT if we used the same retrieval algorithm."

*- the same may be done for the use of aircraft data, e.g. with respect to the transport model uncertainty, and the coverage of aircraft and CONTRAIL measurements and their respective rms differences to provide such information in the paper text since reader's may not go to the discussion documents.*

For the transport model uncertainty, we have expanded the discussion in the following way: "This may be
25 different for previous versions of LMDz: for instance, the refinement of the vertical grid in our version of LMDz from 19 to 39 layers had a major impact (Chevallier et al., 2014). This may be different for or some other off-line models: Schuh et al. (2019), e.g., highlighted regridding problems within one model to explain some large differences with another one."

For the coverage of aircraft and CONTRAIL measurements, we have added the following sentences in section

[revised manuscript text omitted]